# E2ENet: Dynamic Sparse Feature Fusion for Accurate and Efficient 3D Medical Image Segmentation

**Boqian Wu**[1,2,*], **Qiao Xiao**[3,*], **Shiwei Liu**[4], **Lu Yin**[5], **Mykola Pechenizkiy**[3],
**Decebal Constantin Mocanu**[2,3], **Maurice van Keulen**[1], **Elena Mocanu**[1]

[1] University of Twente, [2] University of Luxembourg, [3] Eindhoven University of Technology,
[4] University of Oxford, [5] University of Surrey

{b.wu, m.vankeulen,e.mocanu}@utwente.nl
{q.xiao,m.pechenizkiy}@tue.nl, shiwei.liu@maths.ox.ac.uk,
l.yin@surrey.ac.uk, decebal.mocanu@uni.lu

## Abstract

Deep neural networks have evolved as the leading approach in 3D medical image segmentation due to their outstanding performance. However, the ever-increasing model size and computational cost of deep neural networks have become the primary barriers to deploying them on real-world, resource-limited hardware. To achieve both segmentation accuracy and efficiency, we propose a 3D medical image segmentation model called Efficient to Efficient Network (E2ENet), which incorporates two parametrically and computationally efficient designs. i. Dynamic sparse feature fusion (DSFF) mechanism: it adaptively learns to fuse informative multi-scale features while reducing redundancy. ii. Restricted depth-shift in 3D convolution: it leverages the 3D spatial information while keeping the model and computational complexity as 2D-based methods. We conduct extensive experiments on AMOS, Brain Tumor Segmentation and BTCV Challenge, demonstrating that E2ENet consistently achieves a superior trade-off between accuracy and efficiency than prior arts across various resource constraints. E2ENet achieves comparable accuracy on the large-scale challenge AMOS-CT, while saving over $69\%$ parameter count and $27\%$ FLOPs in the inference phase, compared with the previous best-performing method. Our code has been made available at: https://github.com/boqian333/E2ENet-Medical.

## 1 Introduction

3D medical image segmentation plays an essential role in numerous clinical applications, including computer-aided diagnosis [Yu et al., 2020] and image-guided surgery systems [Ronneberger et al., 2015]. Over the past decade, the rapid development of deep neural networks has achieved tremendous breakthroughs and significantly boosted the progress in this area [Zhou et al., 2018, Huang et al., 2020, Isensee et al., 2021]. However, the model sizes and computational costs are also exploding, which deter their deployment in many real-world applications, especially for 3D models where resource consumption scales cubically [Hu et al., 2021, Valanarasu and Patel, 2022]. This naturally raises a research question: *Can we design a 3D medical image segmentation method that trades off accuracy and efficiency better, subjected to different resource availability?*

Accurately segmenting organs is a challenging task in 3D medical image segmentation due to the variability in size and shape even among the same type of organ, caused by factors such as patient anatomy and disease stage. This diversity amplifies the difficulty to accurately identify and distinguish

38th Conference on Neural Information Processing Systems (NeurIPS 2024).

---

[*]Equal contribution.

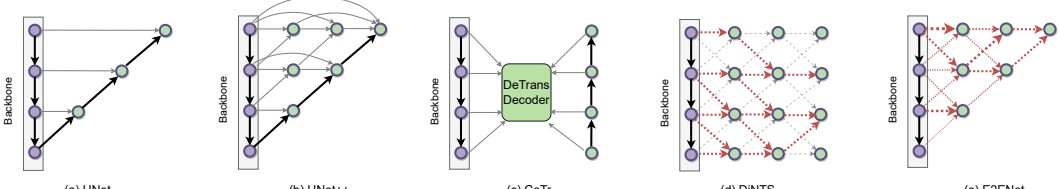

Figure 1: A comparison of feature fusion schemes. The purple nodes depict features extracted from the backbone, while the green nodes depict the fused features. In particular, in DiNTS (d), red lines indicate information flow paths determined through neural architecture search techniques. In E2ENet (e), the red lines with different widths represent sparse information flows determined by the DSFF mechanism, allowing for efficient feature fusion. E2ENet is capable of dynamically learning how many of the features to fuse are derived from the backbone.

the boundaries of different organs, leading to potential errors in segmentation. One of the main solutions for accurate medical image segmentation is to favorably leverage the multi-scale features extracted by the backbone network, but this remains a long-standing problem. Pioneering work UNet [Ronneberger et al., 2015] utilizes skip connection to propagate detailed information to high-level features. More recently, UNet++ [Zhou et al., 2018], CoTr [Xie et al., 2021], and DiNTS [He et al., 2021] have introduced more complex neural network architectures (e.g., dense skip connection and attention mechanism) and optimization techniques (e.g., neural architecture search (NAS) [Elsken et al., 2019]) for cross-scale feature fusion. NAS-based methods search for network topology and operators (e.g., 3x3 Convolution and Maxpool) and subsequently optimize model weights. However, these methods typically require substantial computational resources to explore network topologies and evaluate numerous candidate architectures, making them time-consuming. For instance, C2FNAS [Yu et al., 2020] requires nearly one GPU-year to discover a single 3D segmentation architecture, while DiNTS [He et al., 2021] improves search efficiency but still needs 5.8 GPU days to find a single architecture. In contrast to NAS approaches, our method does not require costly architecture search time and instead directly optimizes and searches for sparse topologies within the predefined architecture.

In this paper, we propose the **Efficient to Efficient Network (E2ENet)**, a model that efficiently incorporates both bottom-up and top-down features from the backbone network in a dynamically sparse pattern, achieving an improved accuracy-efficiency trade-off. As shown in Figure 1 (e), E2ENet incorporates multi-scale features from the backbone into the final output by gradually fusing adjacent features, allowing the network to fully utilize information across various scales. To prevent unnecessary information aggregation, we propose a **dynamic sparse feature fusion (DSFF)** mechanism and embed it in each fusion node. The DSFF mechanism adaptively integrates informative multi-scale features and filters out unnecessary ones during the course of the training process, significantly reducing the computational overhead without sacrificing performance. Additionally, to further improve efficiency, our E2ENet employs a **restricted depth-shift** strategy within 3D convolutions, derived from the temporal shift [Lin et al., 2019] and 3D-shift [Fan et al., 2020] strategies used in efficient video action recognition. This allows the 3D convolution operation with a kernel size of $(1, 3, 3)$ to capture 3D spatial relationships while maintaining the parameter complexity of 2D convolutions and reducing computational cost. To evaluate the performance of our proposed E2ENet, we conducted extensive experiments on the AMOS-CT [Ji et al., 2022], Brain Tumor Segmentation in the Medical Segmentation Decathlon (MSD) [Antonelli et al., 2022], and Multi-Atlas Labeling Beyond the Cranial Vault (BTCV) [Landman et al., 2015] challenges. We found that E2ENet can effectively trade-off between segmentation accuracy and efficiency compared to both convolution-based and transformer-based architectures. In particular, on the AMOS-CT challenge, E2ENet achieves competitive accuracy with a mDice of 90.0%, while being 69% smaller and using 27% fewer FLOPs during inference. With the DSFF mechanism filtering out 90% of feature connections, E2ENet further reduces resource costs without significantly sacrificing accuracy.

## 2 Methodology

In this section, we first present the overall architecture of E2ENet, which allows for the fusion of multi-scale features from three directions. Next, we describe the proposed DSFF mechanism, which adaptively selects informative features and filters out unnecessary ones during training. Lastly, to further increase efficiency, we introduce the use of restricted depth-shift in 3D convolution.

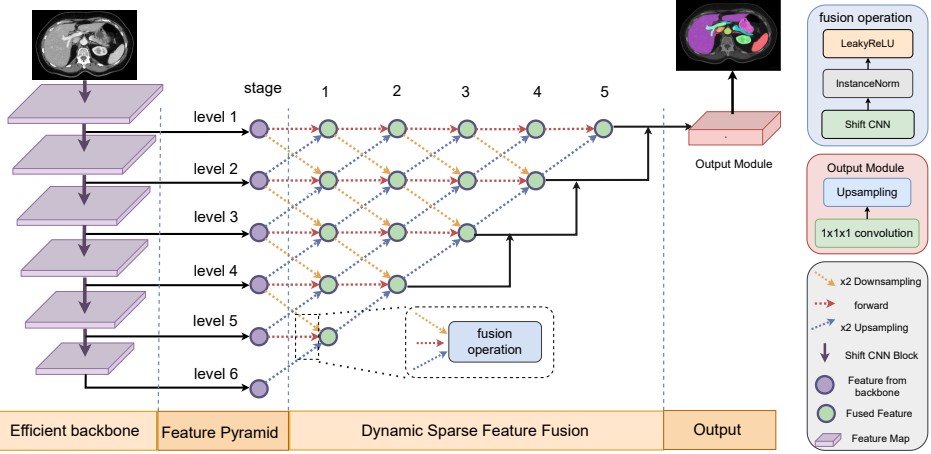

Figure 2: The overall architecture of the proposed E2ENet consists of a CNN backbone that extracts multiple levels of features. These features are then gradually aggregated through several stages, during which the multi-scale features are fused using a fusion operation.

## 2.1 The Architecture

Figure 2 provides an overview of the proposed architecture. Given the input 3D image $I_{in} \in \mathbb{R}^{D \times H \times W}$, the CNN backbone extracts feature maps at multiple scales, represented by $\mathbf{x}^0 = (\mathbf{x}^{0,1}, \mathbf{x}^{0,2}, \ldots, \mathbf{x}^{0,L})$, where $L$ is the total number of feature scales. The feature at level $i$, $\mathbf{x}^{0,i}$ is a tensor with dimensions $d_i \times h_i \times w_i \times c_i$, where $d_i, h_i, w_i$ and $c_i$ represent the depth, height, width, and number of channels of the feature maps at level $i$, respectively. It is worth mentioning that the spatial resolution of the feature maps decreases as the level increases, while the number of channels increases up to a maximum of 320. To fully exploit the hierarchical features extracted by the CNN backbone, these features are aggregated across multiple stages, as depicted in Figure 2 (Dynamic Sparse Feature Fusion section). At stage $j$ ($0 < j \leq L - 1$), the features at level $i$ are obtained by fusing the adjacent features from the previous stage along three directions:

1. "*Downward flow*" (in yellow): The high-resolution feature $\mathbf{x}^{j-1,i-1}$, which provides richer visual details, is passed downward; 2. "*Upward flow*" (in blue): The low-resolution feature $\mathbf{x}^{j-1,i+1}$, which captures more global context, is passed upward; 3. "*Forward flow*" (in red): The features $x^{j-1,i}$, which maintain their spatial resolution, are passed forward for further information integration. The fused cross-scale feature maps at the $i$-th level of the $j$-th stage can be formulated as:

$$\mathbf{x}^{j,i} = \begin{cases} \mathcal{F}^{j,i}([\mathbf{x}^{j-1,1}, \mathcal{U}(\mathbf{x}^{j-1,2})]), & i = 1; \\ \mathcal{F}^{j,i}([\mathcal{D}(\mathbf{x}^{j-1,i-1}), \mathbf{x}^{j-1,i}, \mathcal{U}(\mathbf{x}^{j-1,i+1})]), & others, \end{cases} \quad (1)$$

where $\mathcal{F}^{j,i}(.)$ is a fusion operation, consisting of a convolution operation followed by Instance Normalization (IN) [Ulyanov et al., 2016] and a Leaky Rectified Linear Unit (LeakyReLU) [Maas et al., 2013]. $\mathcal{U}(.)$ and $\mathcal{D}(.)$ denote the up-sampling and down-sampling, respectively. [.] denotes the concatenation operation. In the convolution operation, the input feature maps, which have a channel number of $C_{in}^{j,i}$ ($C_{in}^{j,i} = c_{i-1} + c_i + c_{i+1}$ or $c_i + c_{i+1}$), are fused and processed to produce output feature maps with a channel number of $C_{out}^{j,i}$ ($C_{out}^{j,i} = c_i$). Unlike the UNet++ model, which only considers bottom-up information flows for image segmentation, our proposed E2ENet architecture incorporates both bottom-up and top-down information flows. This allows E2ENet to **make use of both high-level contextual information and fine-grained details** in order to produce more accurate segmentation maps (experimental results can be found in Table 2).

## 2.2 Dynamic Sparse Feature Fusion

Such multi-stage cross-level feature propagation provides a more comprehensive understanding of the images but can also introduce redundant information, necessitating careful handling in feature fusion to ensure efficient interaction between scales or stages. Our proposed Dynamic Sparse Feature Fusion

(DSFF) mechanism addresses the issue of multi-scale feature fusion in an intuitive and effective way. It optimizes the process by enabling **selective and adaptive use of features from different levels during training**. This results in a more efficient feature fusion process with lower computational and memory overhead.

The DSFF mechanism is applied in each fusion operation, allowing the fusion operation $\mathcal{F}^{j,i}(\cdot)$ to select the informative feature maps from the input fused features. The feature map selection process is controlled by a binary mask $\mathbf{M}^{j,i}(\cdot) \in \{0,1\}^{C_{in}^{j,i} \times C_{out}^{j,i}}$, which are trained to filter out $S$ unnecessary feature map connections by zeroing out the corresponding kernels in $\mathcal{F}^{j,i}(\cdot)$. With the DSFF mechanism, the output of the fusion operation is then computed as:

$$\mathbf{x}_{c_{out},:,:,:}^{j,i} = \sigma(\mathcal{I}(\sum_{c=0}^{C_{in}^{j,i}}(\tilde{\mathbf{x}}_{c,:,:,:}^{j,i} * (\mathbf{M}_{c,c_{out}}^{j,i} \cdot \theta_{c,c_{out},:,:,:}^{j,i})))), \tag{2}$$

where $\tilde{\mathbf{x}}_{c,:,:,:}^{j,i}$ is the input fused feature map at the $c$-th channel. $\theta_{c,c_{out},:,:,:}^{j,i}$ is the kernel (feature map connection, as in Figure 3) that connects the $c$-th input feature map to $c_{out}$-th output feature map, $*$ is a convolution operation, $\cdot$ is the product of a scalar (i.e., $\mathbf{M}_{c,c_{out}}^{j,i}$) and a matrix (i.e., $\theta_{c,c_{out},:,:,:}^{j,i}$) [2].

$\mathbf{M}_{c,c_{out}}^{j,i}$ denotes the existence or absence of a connection between the $c$-th input feature map and the $c_{out}$-th output feature map. $\mathcal{I}(.)$ and $\sigma(.)$ denote the Instance Normalization and LeakyReLU, respectively. Thus, the core of DSFF mechanism is the learning of binary masks during the training. At initialization, each binary mask is initialized randomly, with the number of non-zero entries $\|\mathbf{M}^{j,i}\|_0$ equals $(1 - S) \times C_{in}^{j,i} \times C_{out}^{j,i}$. Here, $S$ $(0 < S < 1)$ is a hyperparameter called the feature sparsity level, which determines the percentage of feature map connections that are inactivated. The activated connections can be updated throughout the course of training, while the feature sparsity level $S$ remains constant. This enables efficiency in both testing and training, as the exploitation of connections remains sparse throughout the training process.

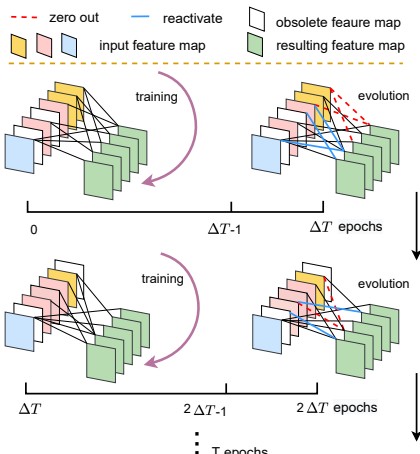

**Every $\Delta T$ training epoch, the activated connections with lower importance are removed, while the same number of deactivated connections are randomly reactivated**, as shown in Figure 3. The importance of activated connection is determined by the $L_1$ norm of the corresponding kernel. That is, for $\mathbf{M}^{j,i}$, the importance score of connection between $c_{in}$-th input feature and $c_{out}$-th output is $\|\theta_{c_{in},c_{out},:,:,:}^{j,i}\|_1$. Our intuition is simple: if a feature map connection is more important (i.e., has a larger effect on output accuracy) than others, the $L_1$ norm of the corresponding kernel should be larger. However, the importance score is suboptimal during training, as the kernels are gradually optimized. Randomly reactivating previous "obsolete" connections with re-initialization can avoid un-

Figure 3: Illustration of our Dynamic Sparse Feature Fusion (DSFF) mechanism. The fusion operation starts from sparse feature connections and allows the connectivity to be evolved after training for $\Delta T$ epochs. During each evolution stage, a fraction of kernels with smaller $L_1$ norms will be zeroed out (red dotted line), while the same fraction of other inactivated connections will be reactivated randomly, keeping the feature sparsity $S$ constant during training (blue solid line).

recoverable feature map abandonment and thoroughly explore the representative feature maps that contribute to the final performance.

The DSFF mechanism allows for the exploration and exploitation of multi-scaled sparse features through a sparse-to-sparse training method. Our method selects features at multiple levels, as opposed to relying solely on feature selection at the input layer [Sokar et al., 2022, Atashgahi et al., 2022]. Additionally, it is a plastic approach to fuse features that can adapt to changing conditions

---

[2]The computation cost of the masking operation is negligible. For instance, consider the feature map $\mathbf{x}_{c,:,:,:}^{j,i}$ with size $D \times H \times W$, the kernel $\theta_{c,c_{out},:,:,:}^{j,i}$ with size of $3 \times 3 \times 3$, and the number of kernels as $C_{in} \times C_{out}$. The computation cost of the masking operation alone is $C_{in} \times C_{out}$, whereas the combined cost of the masking operation and convolution operation is $D \times H \times W \times 3 \times 3 \times 3 \times C_{in} \times C_{out}$.

during training, as opposed to static methods that rely on one-shot feature selection, as shown in the comparison of dynamic and static sparsity in Table 2 of [Mostafa and Wang, 2019]. More details of the training process are elaborated in the Algorithm 1 in Appendix A.3.

## 2.3 Restricted Depth-Shift in 3D Convolution

In 3D medical image segmentation, the 2D-based methods (such as the 2D nnUNet [Isensee et al., 2021]), which apply the 2D convolution to each slice of the 3D image, are computationally efficient but can not fully capture the relationships between slices. To overcome this limitation, we take inspiration from the temporal-shift [Lin et al., 2019] and 3D-shift [Fan et al., 2020] in efficient video action recognition, and the axial-shift [Lian et al., 2021] in efficient MLP architecture for vision. Our proposed E2ENet incorporates a depth-shift strategy in 3D convolution operations, which facilitates inter-slice information exchange and **captures 3D spatial relationships while retaining the simplicity and computational efficiency of 2D-based method**. Temporal-shift [Lin et al., 2019] requires selecting a Shift Proportion (the proportion of channels to conduct temporal shift). Axial-shift [Lian et al., 2021] and 3D-Shift optimize the learnable 3D spatiotemporal shift. We have made refinements to the channel shifting technique by shifting along the depth dimension and incorporating constraints on the shift size. This adaptation is thoughtfully designed to align with the distinctive needs of sparse models employed in medical image segmentation.

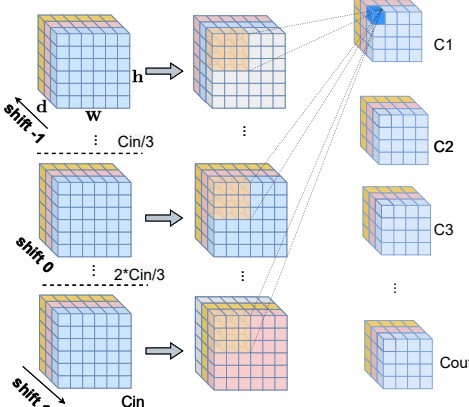

Figure 4: Illustration of restricted depth-shift in 3D Convolution of our E2ENet. The input features (left) are firstly split into 3 parts along the channel dimension, and then shifted by $\{-1, 0, 1\}$ units along the depth dimension respectively (middle). After that, 3D CNNs with kernel size $1 \times 3 \times 3$ are performed on the feature maps (middle) to generate the output features (right).

In our method, we employ a simple depth shift technique by shifting all channels while constraining the shift size to be either +1, 0 or -1, as shown in Figure 4. This choice is motivated using dynamic sparse feature fusion, where the feature maps contain sparse information. If the shift magnitude is too large, it can result in an insufficient representation of channels or an excessive representation of depth information, which can have a negative impact on the effectiveness of the shift operation (experimental results can be found in Table 3).

## 3 Experiments

In this section, we compare the performance of our E2ENet model to baseline methods and report results in terms of both segmentation quality and efficiency. In addition, we will perform ablation studies to investigate the behavior of each component in the E2ENet model. To further analyze the performance of our model, we will examine its generalizability and the effect of model capacity, and present qualitative results by visualizing the predicted segmentations on sample images. We will also visualize the feature fusion ratios to gain insights into which features play an important role in the segmentation process. The description of the dataset and experimental setup can be found in Appendix A.4 and A.5.

Table 1: Quantitative comparisons of segmentation performance on the validation set of AMOS-CT dataset. * denotes the results with postprocessing.

| Methods | mDice (%) ↑ | Params (M) ↓ | FLOPs [1] (G) ↓ | PT score ↑ | mNSD (%) ↑ |
|---|---|---|---|---|---|
| CoTr | 77.1 | 41.87 | 1510.53 | 1.07 | 64.2 |
| nnFormer | 85.6 | 150.14 | 1343.65 | 1.12 | 74.2 |
| UNETR | 78.3 | 93.02 | **391.03** | 1.41 | 61.5 |
| Swin UNETR | 86.4 | 62.83 | 1562.99 | 1.14 | 75.3 |
| VNet | 82.0 | 45.65 | 1737.57 | 1.10 | 67.9 |
| nnUNet | 90.0 | 30.76 | 1067.89 | 1.30 | 82.1 |
| nnUNet* | **90.5** | 30.76 | 1067.89 | 1.31 | **83.0** |
| E2ENet* (s=0.7) | 90.3 | 11.23 | 969.32 | 1.54 | 82.7 |
| E2ENet* (s=0.8) | 90.3 | 9.44 | 778.74 | 1.65 | 82.5 |
| E2ENet* (s=0.9) | 89.9 | **7.64** | 492.29 | **1.89** | 81.8 |
| E2ENet (s=0.7) | 90.1 | 11.23 | 969.32 | 1.54 | 82.3 |
| E2ENet (s=0.8) | 90.0 | 9.44 | 778.74 | 1.65 | 82.3 |
| E2ENet (s=0.9) | 89.6 | **7.64** | 492.29 | 1.88 | 81.4 |

[1] The inference FLOPs are calculated based on the patch sizes of $1 \times 128 \times 128 \times 128$ without considering postprocessing cost.

## 3.1 Comparison with SOTA methods

**AMOS-CT Challenge.** To comprehensively validate our method, we compare it to several state-of-the-art CNN-based models (e.g. nnUNet [Isensee et al., 2021], and Vnet [Milletari et al., 2016]) and transformer-based models (e.g. CoTr [Xie et al., 2021], nnFormer [Zhou et al., 2021], UNETR [Hatamizadeh et al., 2022], and Swin UNETR [Tang et al., 2022]). We record the mDice (class-wise Dice can be found in Table 10 of Appendix), Params, inference FLOPs, and PT score [3] on the validation set [4] of the AMOS-CT challenge in Table 1. E2ENet with a feature sparsity (s) of $0.8$ achieves comparable performance, with a mDice of $90.3\%$, while being $3\times$ smaller in model size and requiring less computational cost during the inference phase compared to the top-performing lightweight model, nnUNet. As the feature sparsity of E2ENet increases, the number of model parameters and inference FLOPs can be further reduced without significantly compromising segmentation performance. This indicates that there is potential to trade-off performance and efficiency by adjusting the feature sparsity of E2ENet. We also report the results of mean normalized surface dice (mNSD), the official segmentation metric for the AMOS challenge, to provide supplementary information on boundary segmentation quality.

**BraTS Challenge in MSD.** E2ENet demonstrates superior performance in terms of mDice compared to other state-of-the-art methods (DiNTS [He et al., 2021], UNet++ [Zhou et al., 2018] and nnUNet [Isensee et al., 2021]). Additionally, it is a resource-efficient network that is competitive with the baselines, as evidenced by its small model size and low inference FLOPs. Specifically, E2ENet model a feature sparsity level of $90\%$ has only 7.63 M parameters, which is significantly smaller than other models yet still outperforms them in terms of

Table 2: 5-fold cross-validation of segmentation performance on the BraTS Challenge training set in the MSD. Note: ED, ET, and NET denote edema, enhancing tumor, and non-enhancing tumor, respectively.

| Methods | Dice (%) ↑ | | | mDice (%) ↑ | Params (M) ↓ | FLOPs [1] (G) ↓ | PT score ↑ |
|---|---|---|---|---|---|---|---|
| | ED | ET | NET | | | | |
| DiNTS | 80.2 | 61.1 | 77.6 | 73.0 | / | / | / |
| UNet++ | 80.5 | 62.5 | 79.2 | 74.1 | 58.38 | 3938.25 | 1.12 |
| nnUNet | 81.0 | 62.0 | 79.3 | 74.1 | 31.20 | 1076.62 | 1.35 |
| E2ENet ($S = 0.7$) | **81.2** | **62.7** | **79.5** | **74.5** | 11.24 | 1067.06 | 1.57 |
| E2ENet ($S = 0.8$) | 81.0 | 62.5 | 79.0 | 74.2 | 9.44 | 780.97 | 1.72 |
| E2ENet ($S = 0.9$) | 80.9 | 62.5 | 79.4 | 74.3 | **7.63** | **494.52** | **2.00** |

[1] The inference FLOPs are calculated based on patch sizes of $4 \times 128 \times 128 \times 128$. The number of parameters and inference FLOPs for DiNTS are not reported because calculating them is time-consuming. This is due to the fact that the architecture for DiNTS is not readily available for the dataset and must be found using Neural Architecture Search (NAS).

mDice. The results on the BraTS dataset, which are MRI images, further validate the effectiveness and efficiency of E2ENet across both CT and MRI medical image analysis.

Due to the limited space and the similarity between the BTCV and AMOS-CT challenges, both of which involve multi-organ segmentation in CT images, the results of the **BTCV challenge** are provided in Appendix A.8.2.

## 3.2 Ablation Studies

In this section, we investigate the impact of two factors on the performance of E2ENet: (i) the DSFF mechanism and (ii) restricted depth-shift in 3D convolution. Specifically, we consider the following scenarios: (#1) w/ DSFF: DSFF mechanism is used to dynamically activate feature map, otherwise, all feature maps are activated during training; (#2) w/ shift: restricted depth/height/width shifting is applied on feature maps before the convolution operation, otherwise, the convolution is performed directly as a standard 3D convolution without shifting operation.

Table 3: Ablation study on the effects of the DSFF mechanism and restricted depth-shift in 3D convolution on the validation set of the AMOS-CT Challenge.

| w/ DSFF | $\Delta T$ (# iters) | w/ shift | shift size | kernel size | mDice (%) ↑ | Params (M) ↓ | FLOPs (G) ↓ | mNSD (%) ↑ |
|---|---|---|---|---|---|---|---|---|
| ✗ | / | ✔ | $(-1,0,1)$ | $(1 \times 3 \times 3)$ | **90.2** | 23.90 | 3069.55 | 82.6 |
| ✗ | / | ✗ | / | $(1 \times 3 \times 3)$ | 88.6 | 23.90 | 3069.55 | 78.6 |
| ✔ | 1200 | ✔ | $(-1,0,1)$ | $(1 \times 3 \times 3)$ | 90.1 | 11.23 | 969.32 | 82.3 |
| ✔ | 1200 | ✗ | / | $(1 \times 3 \times 3)$ | 88.2 | 11.23 | 969.32 | 79.4 |
| ✔ | 1200 | ✔ | $(-2,0,2)$ | $(1 \times 3 \times 3)$ | 89.8 | 11.23 | 969.32 | 82.0 |
| ✔ | 1200 | ✔ | $(-3,0,3)$ | $(1 \times 3 \times 3)$ | 89.7 | 11.23 | 969.32 | 81.6 |
| ✔ | 1200 | ✔ | $(-7,0,7)$ | $(1 \times 3 \times 3)$ | 87.6 | 11.23 | 969.32 | 77.5 |
| ✗ | 1200 | ✗ | / | $(3 \times 3 \times 3)$ | 90.2 | 52.54 | 4511.57 | 82.5 |
| ✔ | 1200 | ✗ | / | $(3 \times 3 \times 3)$ | 90.1 | 27.97 | 1778.55 | 82.1 |
| ✔ | 600 | ✔ | $(-1,0,1)$ | $(1 \times 3 \times 3)$ | 90.0 | 11.23 | 969.32 | 82.2 |
| ✔ | 1800 | ✔ | $(-1,0,1)$ | $(1 \times 3 \times 3)$ | 89.9 | 11.23 | 969.32 | 82.0 |

standard 3D convolution without shifting operation.

---

[3] The explanation of metrics can be found in Appendix A.7.

[4] The validation set includes 100 images and was the previous test set for the first stage of the challenge. In the training of E2ENet, we did not use the validation set and treated it as the test set.

**Effect of DSFF Mechanism.** Table 3 (rows 1 and 3) demonstrates that the E2ENet with the DSFF mechanism, achieved comparable performance while using three times fewer parameters and inference FLOPs. Table 4 shows that removing the DSFF mechanism caused the mDice score to drop from 74.5 % (row 4) to 74.1 % (row 2) on the BraTS challenge, while also increasing the number of parameters by more than $2\times$ and the inference FLOPs by nearly $3\times$. These ablation study results highlight that dynamic sparse feature fusion (DSFF) helps reduce resource costs while maintaining segmentation performance.

To further investigate the impact of the DSFF mechanism, we compared E2ENet with other multi-scale medical image segmentation methods on the AMOS-CT validation dataset, as shown in Table 5. These methods include DeepLabv3 [Chen et al., 2017], CoTr [Xie et al., 2021], and MedFormer [Gao et al., 2022]. DeepLabv3 uses atrous convolution to capture multi-scale context, CoTr integrates multi-scale features using attention, and MedFormer employs all-to-all attention for comprehensive multi-scale fusion, addressing both semantic and spatial aspects.

Table 4: Ablation study on the effects of the DSFF mechanism and restricted depth-shift in 3D convolution, evaluated through 5-fold cross-validation of segmentation performance on the BraTS Challenge training set in the MSD.

| w/ DSFF | w/ shift | kernel size | Dice (%) ↑ ED | Dice (%) ↑ ET | Dice (%) ↑ NET | mDice (%) ↑ | Params (M) ↓ | FLOPs (G) ↓ |
|---|---|---|---|---|---|---|---|---|
| ✗ | ✗ | $(1 \times 3 \times 3)$ | 80.4 | 62.4 | 79.0 | 73.9 | 23.89 | 3071.78 |
| ✗ | ✔ | $(1 \times 3 \times 3)$ | 81.0 | 62.3 | 79.0 | 74.1 | 23.89 | 3071.78 |
| ✔ | ✗ | $(1 \times 3 \times 3)$ | 80.3 | _62.5_ | 79.0 | 73.9 | 11.24 | 1067.06 |
| ✔ | ✔ | $(1 \times 3 \times 3)$ | **81.2** | **62.7** | **79.5** | **74.5** | 11.24 | 1067.06 |
| ✗ | ✗ | $(3 \times 3 \times 3)$ | 80.9 | 61.9 | 79.1 | 74.0 | 52.55 | 4519.26 |
| ✔ | ✗ | $(3 \times 3 \times 3)$ | 81.0 | 62.2 | 79.1 | 74.1 | 28.02 | 2023.52 |
| ✔ | ✗ | $(3 \times 1 \times 3)$ | 80.3 | 61.7 | 78.7 | 73.6 | 11.24 | 1067.06 |
| ✔ | ✗ | $(3 \times 3 \times 1)$ | 80.5 | 61.9 | 78.7 | 73.7 | 11.24 | 1067.06 |
| ✔ | ✔ | $(3 \times 1 \times 3)$ | _81.1_ | 62.4 | 78.9 | 74.1 | 11.24 | 1067.06 |
| ✔ | ✔ | $(3 \times 3 \times 1)$ | 81.0 | 62.3 | _79.4_ | _74.2_ | 11.24 | 1067.06 |

We observed that with a sparsity ratio of 0.9, E2ENet outperforms DeepLabv3 in terms of mDice while significantly reducing computational and memory costs by more than $3\times$ and nearly $10\times$, respectively. Additionally, E2ENet with a sparsity ratio of 0.8 matches MedFormer's performance while significantly reducing computational and memory costs by $3\times$ and $4\times$, respectively. This demonstrates the benefits of multi-scale feature aggregation for medical image segmentation, with E2ENet's DSFF mechanism being much more efficient than other multi-scale methods.

Moreover, as shown in Table 2, E2ENet with the DSFF mechanism outperforms other feature fusion architectures, such as UNet++ and DiNTS, on the BraTS challenge. Overall, the results from the BraTS and AMOS-CT challenges demonstrate that the DSFF mechanism provides an effective and efficient approach for feature fusion.

Finally, we also studied the impact of topology update frequency ($\Delta T$) in the DSFF mechanism and observed that our algorithm is not sensitive to the hyperparameter $\Delta T$.

**Effect of Restricted Depth-Shift in 3D Convolution.** We evaluated the effectiveness of our restricted shift strategy by comparing the performance of E2ENet and its variants. Without restricted depth-shift, the mDice decreased from 74.5% (row 4) to 73.9% (row 3) for the BraTS challenge as shown in Table 4, and decreased from 90.1% (row 4) to 88.2% (row 3) for AMOS-CT (as shown in Table 3). In Table 3, when comparing E2ENet with kernel sizes of 3x3x3

Table 5: Comparison with other multi-scale medical image segmentation methods on the AMOS-CT validation dataset. The mDice score for MedFormer is sourced from [Gao et al., 2022].

| method | mDice (%) ↑ | Params (M) ↓ | FLOPs (G) ↓ |
|---|---|---|---|
| DeepLabv3 | 89.0 | 74.68 | 1546.25 |
| CoTr | 77.1 | 41.87 | 1510.53 |
| MedFormer | **90.1** | 39.59 | 2332.75 |
| E2ENet (S=0.8) | _90.0_ | 9.44 | 778.74 |
| E2ENet (S=0.9) | 89.6 | **7.64** | **492.29** |

(rows 8 and 9) to E2ENet with kernel sizes of 1x3x3 combined with a depth shift (rows 1 and 3), their segmentation accuracy remains the same, whether DSFF is used or not. In Table 4, we find that E2ENet with kernel sizes of 1x3x3 combined with a depth shift (rows 2 and 4) even improves segmentation accuracy, whether without DSFF or with DSFF, when compared to kernel sizes of 3x3x3 (rows 5 and 6). This further demonstrates that our proposed efficient Restricted Depth-Shift 3D Convolutional layer, which utilizes a 1x3x3 kernel with restricted depth shift, is equivalent to a 3x3x3 kernel in terms of segmentation accuracy. Moreover, it offers significant savings in computational and memory resources. This demonstrates that the use of a 1x3x3 kernel with restricted depth shift is functionally equivalent to a 3x3x3 kernel in terms of segmentation accuracy, all the while offering savings in computational and memory resources.

Rows 9 and 10 of Table 4 demonstrate that when restricted shift is applied to the height or width dimensions, the model's performance decreases compared to E2ENet (4th row) with restricted shift on the depth dimension, called as restricted depth-shift. We also observed that the performance of 3D convolution with kernel sizes of $1 \times 3 \times 3$ (row 3), $3 \times 1 \times 3$ (row 7), and $3 \times 3 \times 1$ (row 8) decreased significantly without restricted shift compared to their counterparts with restricted shift on the corresponding dimensions (rows 4, 9, and 10, respectively). These phenomena demonstrate the effectiveness of restricted shift, especially restricted depth-shift, for medical image segmentation.

Furthermore, we evaluated the impact of different shift sizes on the model's performance. As shown in Table 3, we compared the performance of E2ENet with shift sizes of (-1, 0, 1) to (-2, 0, 2), (-3, 0, 3) and (-7, 0, 7), and observed a decrease in performance from $90.1\%$ to $89.8\%$, $89.7\%$ and $87.6\%$, respectively, as the shift size increased. Increasing the shift size means considering more depth-wise information at the expense of channel-wise information, leading to an insufficient representation of channels, as discussed in Section 2.3. Additionally, a large shift size (equivalent to a large kernel size) leads to a loss of local spatial relationships, which are crucial for segmentation. This results in a blurring effect that reduces the precision of boundary alignment, particularly affecting metrics like mNSD, which rely heavily on accurate boundary information. Therefore, we use a shift size of (-1, 0, 1) in our restricted depth-shift strategy as the default setting in our experiments. Finally, we plot a critical distance diagram to show the statistical significance of our modules in Appendix A.8.3.

Table 6: Evaluating the generalizability of E2ENet on the AMOS-MRI dataset. CT → MRI: pretrained on the CT dataset and fine-tuned on the MRI dataset; MRI: trained solely on the MRI dataset; CT+MRI: trained on both the CT and MRI datasets. Results for all models, except E2ENet and nnUNet (CT → MRI), are sourced from the AMOS website.

| method | mDice (%) ↑ | | | mNSD (%) ↑ | | |
|---|---|---|---|---|---|---|
| | CT → MRI | MRI | CT + MRI | CT → MRI | MRI | CT + MRI |
| E2ENet | **87.8** | / | / | **83.9** | / | / |
| nnUNet | 87.4 | 87.1 | 87.7 | 83.3 | 83.1 | 82.7 |
| VNet | / | 83.9 | 75.4 | / | 65.8 | 65.6 |
| nnFormer | / | 80.6 | 75.48 | / | 74.0 | 66.63 |
| CoTr | / | 77.5 | 73.46 | / | 78.0 | 65.35 |
| Swin UNETR | / | 75.7 | 77.52 | / | 65.3 | 69.10 |
| UNETR | / | 75.3 | / | / | 70.1 | / |

## 3.3 Generalizability Analysis

To evaluate the generalizability of the resulting architectures, we compared our model, pre-trained on AMOS-CT and fine-tuned on AMOS-MRI, with nnUNet, which was pre-trained on AMOS-CT and fine-tuned on AMOS-MRI, and both models are fine-tuned for 250 epochs. It is worth noting that the topology (weight connections) is determined through AMOS-CT pre-training and remains fixed during the fine-tuning phase. From Table 6, we discovered that the E2ENet architecture, initially designed for AMOS-CT, can be effectively applied to AMOS-MRI as well. This adaptation resulted in improved performance compared to nnUNet and other baselines. nnUNet

Table 7: Quantitative comparisons of segmentation performance on AMOS-CT test dataset. ‡ and † denote the results with and without postprocessing that are reproduced by us. * indicates the results with postprocessing.

| Methods | mDice (%) ↑ | Params (M) ↓ | FLOPs (G) ↓ | PT score ↑ | mNSD (%) ↑ |
|---|---|---|---|---|---|
| CoTr | 80.9 | 41.87 | 1510.53 | 1.11 | 66.3 |
| nnFormer | 85.6 | 150.14 | 1343.65 | 1.11 | 72.5 |
| UNETR | 79.4 | 93.02 | **391.03** | 1.41 | 60.8 |
| Swin-UNETR | 86.3 | 62.83 | 1562.99 | 1.13 | 73.8 |
| VNet | 82.9 | 45.65 | 1737.57 | 1.11 | 67.6 |
| nnUNet† | 90.6 | 30.76 | 1067.89 | 1.31 | 82.0 |
| nnUNet‡ | **91.0** | 30.76 | 1067.89 | 1.31 | 82.6 |
| E2ENet* (s=0.7) | 90.7 | 11.23 | 969.32 | 1.54 | 82.2 |
| E2ENet* (s=0.8) | 90.7 | 9.44 | 778.74 | 1.65 | 82.1 |
| E2ENet* (s=0.9) | 90.4 | **7.64** | 492.29 | **1.89** | 81.4 |
| E2ENet (s=0.7) | 90.6 | 11.23 | 969.32 | 1.54 | 82.0 |
| E2ENet (s=0.8) | 90.4 | 9.44 | 778.74 | 1.65 | 81.8 |
| E2ENet (s=0.9) | 90.1 | **7.64** | 492.29 | **1.89** | 80.7 |

and these baselines were either transferred from CT, trained solely on MRI data, or jointly on MRI and CT datasets.

In the AMOS-CT dataset, there is a domain shift between the training and test datasets due to variations in the image acquisition scanners [Ji et al., 2022]. Thus, we also assess the generalizability of our approach by evaluating its performance on the AMOS-CT test dataset. As demonstrated in Table 7, E2ENet exhibits comparable performance even in the presence of domain shift when compared to nnUNet. This is achieved while maintaining lower computational and memory costs.

## 3.4 Model Capacity Analysis

To ensure a fair comparison, we scale down nnUNet by reducing the number of channels at level 1 from 32 to 27 and decreasing the total number of feature scales from 5 to 4, resulting in a modified version referred to as nnUNet (-). Additionally, we scale up E2ENet by increasing the number of channels at level 1 from 48 to 58, creating a modified version referred to as E2ENet (+). We evaluated the performance of these models, and the results can be found in Table 8. It is worth noting that scaling down nnUNet resulted in decreased performance in terms of mDice and mNSD, while scaling up E2ENet led to an increase in mDice and comparable performance in terms of mNSD. This indicates that the comparable performance of the memory and computation-efficient E2ENet is not attributed to the dataset's requirement for a small number of parameters and computations.

## 3.5 Qualitative Results

In this section, we compare the proposed E2ENet and nnUNet qualitatively on three challenges. To make the comparison easier, we highlight detailed segmentation results with red dashed boxes.

Table 8: A comparison under a similar FLOPs budget on the AMOS-CT challenge in two cases: when nnUNet is scaled down and when E2ENet is scaled up.

| Method | mDice (%) ↑ | Params (M) ↓ | FLOPs (G) ↓ | mNSD (%) ↑ |
|---|---|---|---|---|
| E2ENet (S=0.8) | 90.0 | **9.43** | 778.74 | **82.3** |
| nnUNet | 90.0 | 31.20 | 1067.89 | 82.1 |
| nnUNet (-) | 89.7 | 12.96 | **755.79** | 81.9 |
| E2ENet (+) | **90.4** | 10.37 | 1119.17 | 82.2 |

Figure 5 presents a qualitative comparison of our proposed E2ENet with nnUNet on the AMOS-CT challenge. As a widely used baseline, nnUNet has been evaluated on multiple medical image segmentation challenges. Our results demonstrate that, on certain samples, E2ENet can improve segmentation quality and overcome some of the challenges faced on the AMOS-CT challenge. For example, as shown in the first column, E2ENet more accurately distinguishes between the stomach and the esophagus, which is challenging due to their close proximity. In the second column, E2ENet better differentiates the duodenum from the background, while in the third column, E2ENet accurately identifies the precise boundaries of the liver, a structure that is prone to over-segmentation. These examples demonstrate the potential of our proposed method to improve the accuracy of medical image segmentation to some extent, especially in challenging cases such as distinguishing between closely located organs or accurately segmenting complex shapes. The Qualitative Results of BTCV and BraTS Challenge can be found in Appendix A.9.1.

## 3.6 Feature Fusion Visualization

In this section, we explore how the DSFF mechanism fuses features from three directions (upward, forward, and downward) during training. To shed light on this mechanism, we specifically analyze the proportions [5] of feature map connections from different directions to a specific fused feature node, considering a feature sparsity level of 0.8. After training, we observed in Figure 6 (d) that the DSFF module learned to assign greater importance to features from the "forward" directions for most fused feature nodes. For example, in the first feature level, the flow and processing of features can be observed as if they were passing through a fully convolutional neural network (FCN), which preserves the spatial dimensions of the input image. At the second feature level, the original image is downsampled by a factor of $1/2$ and flows through another FCN. Therefore, from this perspective, E2ENet can take advantage of FCN to effectively incorporate and preserve multi-scale information.

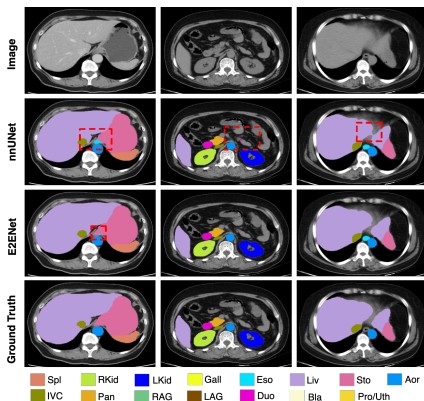

Figure 5: Qualitative comparison of E2ENet and nnUNet on the AMOS-CT challenges.

---

[5]We calculated these proportions by dividing the number of non-zero connections in a given direction by the total number of non-zero connections within each fused feature.

Simultaneously, the complementary feature flows from the 'upward' and 'downward' directions provide richer information. At the later fusion step of the lower feature levels, the 'upward' information becomes more dominant than the 'downward' information. This prioritization of upward feature flow is similar to the design of the decoders in UNet and UNet++. While E2ENet alleviates the semantic gap more effectively, the proportion of feature map connections in the 'downward' direction always has a certain ratio. This also allows the network to better preserve low-level information and integrate it with high-level information, leading to improved performance in capturing fine details. Interestingly, this trend becomes increasingly apparent during training, as illustrated in Figure 6 (a), (c) and (d).

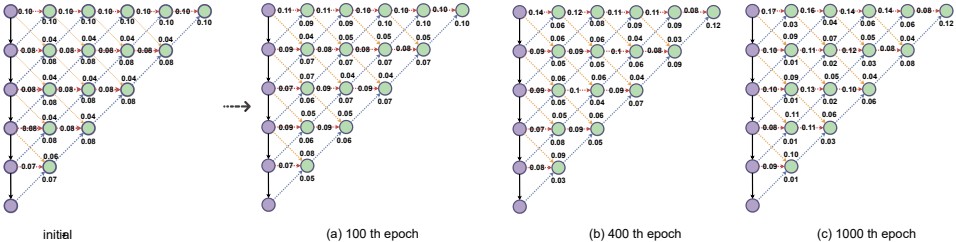

Figure 6: The proportions of feature connections during training with the DSFF mechanism at a feature sparsity level of 0.8 on the AMOS-CT challenge.

# 4 Conclusion

In this work, our aim is to address the challenge of designing a 3D medical image segmentation method that is both accurate and efficient. By proposing a dynamic sparse feature fusion mechanism and incorporating restricted depth-shift in 3D convolution, our E2ENet improves performance on 3D medical image segmentation tasks while significantly reducing computational and memory overhead. The dynamic sparse feature fusion mechanism demonstrates its ability to adaptively learn the importance of each feature map, zeroing out the less important ones. This results in a more efficient feature representation without compromising performance. Additionally, the experiments demonstrate that restricted depth-shift in 3D convolution enables the model to capture spatial information more effectively and efficiently. Extensive experiments on three benchmarks demonstrate that E2ENet consistently achieves a superior trade-off between accuracy and efficiency compared to previous state-of-the-art baselines. While E2ENet provides a promising solution for balancing accuracy and computational cost, future work could explore the potential of a learnable shift offset, which may further improve performance. Additionally, as plug-and-play components, the DSFF mechanism and restricted depth-shift could be interesting to apply to other 3D segmentation models to explore their potential in the future. Furthermore, future advancements in hardware support for sparse neural networks could fully unlock the potential of sparse training methods.

# 5 Acknowledgements

Qiao Xiao is supported by the research program 'MegaMind - Measuring, Gathering, Mining and Integrating Data for Self-management in the Edge of the Electricity System', (partly) financed by the Dutch Research Council (NWO) through the Perspectief program under number P19-25. Elena Mocanu is partly supported by the Modular Integrated Sustainable Datacenter (MISD) project funded by the Dutch Ministry of Economic Affairs and Climate under the European Important Projects of Common European Interest - Cloud Infrastructure and Services (IPCEI-CIS) program for 2024-2029. This research used the Dutch national e-infrastructure with the support of the SURF Cooperative, using grant no. EINF-7499.

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

# A Appendix.

## A.1 Impact Statement

In an era dominated by over-parameterized models, designing resource-aware AI models is becoming increasingly important, especially for time-consuming tasks like medical segmentation. Our insights into model efficiency training have the potential to broaden the application of deep neural networks in this area. Overall, this work advances our fundamental understanding of dynamic sparse training and offers future perspectives for scalable and efficient AI models. We do not anticipate any negative societal impacts resulting from this research.

## A.2 Related Work

### A.2.1 3D Medical Image Segmentation

Convolutional neural networks (CNNs) have become the dominant architecture for 3D medical image segmentation in recent years (e.g. 3D UNet [Çiçek et al., 2016], UNet++ [Zhou et al., 2018], UNet3+ [Huang et al., 2020], PaNN [Zhou et al., 2019] and nnUNet [Isensee et al., 2021]), due to their ability to capture local and weight-sharing dependencies [d'Ascoli et al., 2021, Dai et al., 2021]. However, some recent methods have attempted to incorporate transformer modules into CNNs (e.g. CoTr [Xie et al., 2021], TransBTS [Wang et al., 2021]), or use pure transformer architectures (e.g. ConvIt [Karimi et al., 2021], nnFormer [Zhou et al., 2021], Swin UNet [Cao et al., 2021]), in order to capture long-range dependencies. These transformer-based approaches often require large amounts of training data, longer training times, or specialized training techniques, and can also be computationally expensive. Most recently, a novel architecture called Mamba [Gu and Dao, 2023] has shown potential for computational efficiency as a State Space model in handling long sequences and has been applied to medical image segmentation tasks [Ruan and Xiang, 2024, Xing et al., 2024, Wang et al., 2024]. However, it has led to underwhelming performance compared to state-of-the-art convolutional models. In this paper, we propose an alternative method for efficiently incorporating 3D contextual information using a restricted depth-shift strategy in 3D convolutions, and further improving performance through adaptive multi-scale feature fusion.

### A.2.2 Feature Fusion in Medical Image Segmentation

Multi-scale feature fusion is a crucial technique in medical image segmentation that allows a model to detect objects across a range of scales, while also recovering spatial information that is lost during pooling [Wang et al., 2022, Xie et al., 2021]. However, effectively representing and processing multi-scale hierarchy features can be challenging, and simply summing them up without distinction can lead to semantic gaps and degraded performance [Wang et al., 2022, Tan et al., 2020]. To address this issue, various approaches have been proposed, including adding learnable operations to reduce the gap with residuals [Ibtehaz and Rahman, 2020], attention blocks [Oktay et al., 2018]. More recently, UNet++ [Zhou et al., 2018] and its variants [Li et al., 2020, Huang et al., 2020, Jha et al., 2019] have adapted the gating signal to dense nesting levels, taking into account as many feature levels as possible. NAS-UNet [Weng et al., 2019] tries to automatically search for better feature fusion topology. While these methods have achieved better performance, they can also incur significant computational and information redundancy. Dynamic convolution [Su et al., 2020, Chen et al., 2020] utilizes coefficient prediction or attention modules to dynamically aggregate convolution kernels, thereby reducing computation costs. In our paper, we propose an intuitive approach to optimizing multi-scale feature fusion, which enables selective leveraging of sparse feature representations from fine-grained to semantic levels through the proposed dynamic sparse feature fusion mechanism.

### A.2.3 Sparse Training

Recently, sparse training techniques have shown the possibility of training an efficient network with sparse connections that match (or even outperform) the performance of dense counterparts with lower computational cost [Mocanu et al., 2018, Liu et al., 2021b]. Beginning with [Mocanu et al., 2016], it has been demonstrated that initializing a static sparse network without optimizing its topology during training can also yield comparable performance in certain situations [Lee et al., 2018, Tanaka et al., 2020, Wang et al., 2019]. However, Dynamic Sparse Training (DST), also known as sparse training with dynamic sparsity [Mocanu et al., 2018], offers a different approach by jointly optimizing the

**Algorithm 1** The Training Process of Dynamic Sparse Feature Fusion (DSFF)

---

**Require:** Dataset $\mathcal{X}$ with label $\mathcal{Y}$; feature sparsity $S$; backbone $f_\Theta(.)$; Output Module: $f_{out}$; Total training epochs: $T$;

evolution period: $\Delta T$; connection updating number: $f_{decay}\left(\Delta T; \alpha, T\right) = \frac{\alpha}{2}\left(1 + \cos\left(\frac{\Delta T \pi}{T}\right)\right)$, $\alpha$ represents the number of updated connections during the initial topology update, which is set to $1/2$; Loss function: $\mathcal{L}(.)$; fusion operation: $\mathcal{F}^{j,i}(\cdot)$ with convolution kernels $\theta^{j,i}$, where the numbers of input and output channel are $C_{in}^{j,i}, C_{out}^{j,i}$.

1: $\mathbf{M}^{j,i} \leftarrow$ random initialize masks for all levels and stages, satisfying that $\|\mathbf{M}^{j,i}\|_0$ equals $(1 - S) \times C_{in}^{j,i} \times C_{out}^{j,i}$
2: **for** $t = 1$ **to** $T$ **do**
3:     Sample a batch $I_t, Y_t \sim \mathcal{X}, \mathcal{Y}$
4:     Generate multi-scaled features: $\left(\mathbf{x}^{0,1}, \mathbf{x}^{0,2}, \ldots, \mathbf{x}^{0,L}\right) = f_\Theta(I_t)$
5:     **for** each stage $j = 1$ **to** $L - 1$ **do**
6:         **for** each level $i = 1$ **to** $L - j$ **do**
7:             **if** $i = 1$ **then**
8:                 $\mathbf{x}^{j,i} = \mathcal{F}^{j,i}([\mathbf{x}^{j-1,1}, \mathcal{U}(\mathbf{x}^{j-1,2})])$
9:             **else**
10:                 $\mathbf{x}^{j,i} = \mathcal{F}^{j,i}([\mathcal{D}(\mathbf{x}^{j-1,i-1}), \mathbf{x}^{j-1,i}, \mathcal{U}(\mathbf{x}^{j-1,i+1})])$
11:             **end if**
12:         **end for**
13:     **end for**
14:     $l_t = 4/7\mathcal{L}(f_{out}\left(\mathbf{x}^{L-1,1}\right), Y_i) + 2/7\mathcal{L}(f_{out}\left(\mathbf{x}^{L-2,2}\right), \mathcal{D}(Y_i)) + 1/7\mathcal{L}(f_{out}\left(\mathbf{x}^{L-3,3}\right), \mathcal{D}(\mathcal{D}(Y_i)))$
15:     **if** $(t \bmod \Delta T) == 0$ **then**
16:         **for** each stage $j = 1$ **to** $L - 1$ **do**
17:             **for** each level $i = 1$ **to** $L - j$ **do**
18:                 $u = (C_{in}^{j,i} \times C_{out}^{j,i}) f_{decay}\left(t; \alpha, T\right)\left(1 - S\right)$
19:                 $IS \leftarrow$ importance score ($L_1$ Norm of corresponding kernel) for activated each feature connection
20:                 $\mathbb{I}_{activate} = RandomK(\mathbb{I}_{inactivate}, u)$
21:                 $\mathbb{I}_{inactivate} = ArgTopK\left(-IS, u\right)$
22:                 $\mathbf{M}^{j,i} \leftarrow$ Update $\mathbf{M}^{j,i}$ using $\mathbb{I}_{inactivate}$ and $\mathbb{I}_{activate}$
23:             **end for**
24:         **end for**
25:     **else**
26:         Training the E2ENet using SGD optimizer
27:     **end if**
28: **end for**

---

sparse topology and weights during the training process starting from a sparse network [Liu et al., 2021a, 2022, Evci et al., 2020, Jayakumar et al., 2020, Mostafa and Wang, 2019, Yuan et al., 2021]. This allows the model's sparse connections to gradually evolve in a prune-and-grow scheme, leading to improved performance compared to naively training a static sparse network [Liu et al., 2021c, Xiao et al., 2022]. In contrast to prior methods that aim to find sparse networks that can match the performance of corresponding dense networks, we aim to leverage DST to adaptively fuse multi-scale features in a computationally efficient manner for 3D medical image segmentation.

## A.3   Algorithm

## A.4   Datasets and Experiment Setup

**AMOS-CT:** The Abdominal Multi-Organ Segmentation Challenge (AMOS) [Ji et al., 2022] task 1 consists of 500 computerized tomography (CT) cases, including 200 scans for training, 100 for validation, and 200 for testing. These cases have been collected from a diverse patient population and include annotations of 15 organs. The scans are from multiple centers, vendors, modalities, phases, and diseases.

**BTCV:** The Beyond the Cranial Vault (BTCV) abdomen challenge dataset [6] consists of 30 CT scan images for training and 20 for testing. These images have been annotated by interpreters under the supervision of radiologists, and include labels for 13 organs.

**BraTS:** The Brain Tumor Segmentation Challenge in the Medical Segmentation Decathlon (MSD) [Antonelli et al., 2022, Simpson et al., 2019] consists of 484 MRI images from 19 different institutions. These images contain three different tumor regions of interest (ROIs): edema (ED), non-enhancing tumor (NET) and enhancing tumor (ET). The goal of the challenge is to segment these ROIs in the images accurately.

### A.5 Implementation Details

In our work, we utilized the PyTorch toolkit [Paszke et al., 2019] on an NVIDIA A100 GPU for all our experimental evaluations. We also used the nnUNet codebase [Isensee et al., 2021] to pre-process data before training our proposed E2ENet model. For the AMOS dataset, we used the nnUNet codebase as the benchmark implementation.

For training, we use the stochastic gradient descent (SGD) optimizer with an initial learning rate of 0.01, which is gradually decreased through a "poly" decay schedule. The optimizer is configured with a momentum of 0.99 and a weight decay of $3 \times 10^{-5}$. The maximum number of training epochs is 1000, with 250 iterations per epoch. For the loss function, we combine both cross-entropy loss and Dice loss as in [Isensee et al., 2021]. To improve performance, various data augmentation techniques such as random rotation, scaling, flipping, adding Gaussian noise, blurring, adjusting brightness and contrast, simulating low resolution, and Gamma transformation are used before training.

We employ a 5-fold cross-validation strategy on the training set for all experiments, selecting the final model from each fold and simply averaging their outputs for the final segmentation predictions. In the testing stage, we employ the sliding window strategy, where the window sizes are equal to the size of the training patches. Additionally, post-processing methods outlined in [Isensee et al., 2022] are applied for the AMOS-CT dataset during the testing phase.

### A.6 The Architecture

The backbone generates a total of $L = 6$ multi-scale feature levels, each with a specified number of channels: [c1, c2, c3, c4, c5, c6] = [48, 96, 192, 320, 320, 320]. At each level of feature generation, there are two convolution layers with a kernel size of (1, 3, 3), followed by instance normalization and the application of leaky ReLU activation. The down-sampling ratios for each level are as follows: ((1, 2, 2), (2, 2, 2), (2, 2, 2), (2, 2, 2), (2, 2, 2), (2, 2, 2), (2, 2, 2)).

### A.7 Evaluation Metrics

#### A.7.1 Mean Dice Similarity Coefficient

To assess the quality of the segmentation results, we use the mean Dice similarity coefficient (mDice), which is a widely used metric in medical image segmentation. The mDice is calculated as follows:

$$mDice = \frac{1}{N} \sum_{j=1}^{N} \frac{2|\mathbf{y}_j \cdot \hat{\mathbf{y}}_j|}{(|\mathbf{y}_j| + |\hat{\mathbf{y}}_j|)}, \tag{3}$$

where $N$ is the number of classes, $\cdot$ is the pointwise multiplication, $\mathbf{y}_j$ and $\hat{\mathbf{y}}_j$ represent the ground truth and predicted masks of the $j$-th class, respectively, which are encoded in one-hot format. $\frac{2|\mathbf{y}_j \cdot \hat{\mathbf{y}}_j|}{(|\mathbf{y}_j| + |\hat{\mathbf{y}}_j|)}$ is the Dice of $j$-th class, which measures the overlap between the predicted and ground truth segmentation masks for that class.

#### A.7.2 Number of Parameters

The size of the network can be estimated by summing the number of non-zero parameters (Params), which includes the parameters of activated sparse feature connections (kernels) and parameters of the

---

[6]https://www.synapse.org/#!Synapse:syn3193805/wiki/89480

backbone. The calculation is given by the following equation:

$$Params = \|\Theta\|_0 + \sum_{j=1}^{L-1} \sum_{i=1}^{L-j} \sum_{c_{in}=1}^{C_{in}^{j,i}} \sum_{c_{out}=1}^{C_{out}^{j,i}} \mathbf{M}_{c_{in},c_{out}}^{j,i} \|\theta_{c_{in},c_{out}}^{j,i}\|_0. \tag{4}$$

Here, $\Theta$ is the parameter from backbone, $L$ is the total number of feature levels, $\mathbf{M}^{j,i}$ is a matrix of size $C_{in}^{j,i} \times C_{out}^{j,i}$, and $\mathbf{M}_{c_{in},c_{out}}^{j,i}$ indicates whether the kernel $\theta_{c_{in},c_{out}}^{j,i}$ connecting the $c_{in}$-th input and $c_{out}$-th output feature map exist or not. The $L_0$ norm $\|\theta_{c_{in},c_{out}}^{j,i}\|_0$ provides the number of non-zero entries of $\theta_{c_{in},c_{out}}^{j,i}$.

### A.7.3 Float Point Operations

Floating point operations (FLOPs) is a commonly used metric to compare the computational cost of a sparse model to that of a dense counterpart [Hoefler et al., 2021] [7]. In our comparison, it is calculated by counting the number of multiplications and additions performed in only one forward pass of the inference process without considering postprocessing. The inference FLOPs are estimated layer by layer and depend on the sparsity level of the network. For each convolution or transposed convolution layer, the inference FLOPs is calculated as follows:

$$FLOPs_{conv} = (2K_d K_h K_w C_{in}(1-S) + 1) \times C_{out} HWD, \tag{5}$$

where $K_d$, $K_h$ and $K_w$ are the kernel sizes in depth, height and width; $S$ is the feature sparsity level, for layers that are not part of the DSFF mechanism, $S = 0$ is used; $C_{in}$ and $C_{out}$ are the numbers of input feature and output feature; $H$, $W$ and $D$ are the height, width and depth of output features. For each fully connected layer, the inference FLOPs is calculated as follows:

$$FLOPs_{fc} = (2C_{in}(1-S) + 1) \times C_{out}. \tag{6}$$

### A.7.4 Performance Trade-Off Score

The accuracy-efficiency trade-offs could be further analyzed, from comparing resource requirements to describing holistic behaviours (including mDice, Params and inference FLOPs) for the 3D image segmentation methods. To quantify these trade-offs, we introduce the Performance Trade-Off (PT) score, which is defined as follows:

$$PT = \alpha_1 \frac{mDice}{mDice_{max}} + \alpha_2 \left(\frac{Params_{min}}{Params} + \frac{FLOPs_{min}}{FLOPs}\right), \tag{7}$$

where $\alpha_1$ and $\alpha_2$ are weighting factors, which control the trade-off between accuracy performance and resource requirements, and $mDice_{max}$, $Params_{min}$, and $FLOPs_{min}$ denote the highest mDice score, the smallest number of parameters, and the lowest inference FLOPs among the compared methods for a specific dataset, respectively. The term $\frac{mDice}{mDice_{max}}$ measures the segmentation accuracy, while $\frac{Params_{min}}{Params} + \frac{FLOPs_{min}}{FLOPs}$ measures the resource cost.

In most cases, we consider both segmentation accuracy and resource cost to be equally important, thus we set $\alpha_1 = 1$ and $\alpha_2 = 1/2$ in the following experiments. However, we also explore the impact of different choices of $\alpha_1$ and $\alpha_2$, as detailed in Section A.9. The PT score serves as a valuable metric for evaluating the trade-offs between segmentation accuracy and efficiency.

### A.7.5 Discussion on Running Time

Given the relatively restricted support for sparse operations in current off-the-shelf commodity GPUs and TPUs without sparsity-aware accelerators, we did not attempt to achieve practical speedup during training. Instead, we chose to implement our models with binary masks in our work. The promising benefits of dynamic sparsity presented in this study have not yet translated into actual speedup. Accelerating training time will be a focus for our next work.

---

[7]This is because current sparse training methods often use masks on dense weights to stimulate sparsity. This is done because most deep learning hardware is optimized for dense matrix operations. As a result, using these prototypes doesn't accurately reflect the true memory and speed benefits of a truly sparse network [Hoefler et al., 2021].

Although not the focus of our current work, it would be interesting for future work to examine the speedup results of sparse operation during training, using such specialized hardware accelerators. For example, at high unstructured sparsity levels, XNNPACK [8] has already shown significant speedups over dense baselines on smartphone processors.

Although the support for unstructured sparsity on GPUs remains relatively limited, its practical relevance has been widely demonstrated on non-GPU hardware, such as CPUs and customized accelerators. For example, FPGA accelerators have demonstrated significant acceleration and energy efficiency for unstructured sparse RNNs, outperforming commercial CPUs and GPUs by effectively leveraging the FPGA's embedded resources. Additionally, DeepSparse [9] has shown impressive results in deploying large-scale, BERT-level sparse models on modern Intel CPUs. This approach achieved a 10× compression in model size with under a 1% accuracy loss, a 10× increase in CPU inference speed with less than a 2% accuracy reduction, and an impressive 29× CPU inference speedup with a maximum accuracy drop of 7.5% [Liu and Wang, 2023].

Table 9: Comparison of inference speeds between E2ENet and nnUNet.

| Methods | nnUNet | E2ENet(s=0.8) | E2ENet(s=0.9) |
|---|---|---|---|
| Latency(ms) | 10.07 | 8.02 | 7.28 |
| Throughput(items/sec) | 99.19 | 124.60 | 137.13 |
| speedup | 1.0× | 1.26× | 1.38× |

Inspired by these advancements, we adopted an approach based on DeepSparse. As shown in Table 9, we conducted experiments with patches of images sized 32×32×32 as input, comparing the CPU wall-clock timings for online inference between our proposed E2ENet and nnUNet on an Intel Xeon Platinum 8360Y CPU with 18 cores. We acknowledge that, while our proposed models with sparsity do achieve speedups in practical inference, they are not as pronounced as those observed with BERT-level sparse models. This is primarily due to the nature of segmentation and 3D convolution operations. However, this presents a promising avenue for our future work.

## A.8 More Experimental Results

### A.8.1 Class-wise Dice of AMOS-CT

Table 10: Quantitative comparisons (class-wise Dice (%) ↑, mDice(%)↑, Params(M)↓, inference FLOPs(G)↓, PT score↑ and mNSD(%)↑) of segmentation performance on the validation set of AMOS-CT dataset. **Bold** indicates the best and underline indicates the second best. Note: Spl: spleen, RKid: right kidney, LKid: left kidney, Gall: gallbladder, Eso: esophagus, Liv: liver, Sto: stomach, Aor: aorta IVC: inferior vena cava, Pan: pancreas, RAG: right adrenal gland, LAG: left adrenal gland, Duo: duodenum, Bla: bladder, Pro/Uth: prostate/uterus. The class-wise Dice, mDice and mNSD results of baselines, except for nnUNet, are collected from the [Ji et al., 2022]. [†] indicates the results without postprocessing that are collected from the AMOS website. [‡] denotes the results with postprocessing that are reproduced by us. [*] indicates the results with postprocessing.

| Methods | Spl | RKid | LKid | Gall | Eso | Liv | Sto | Aor | IVC | Pan | RAG | LAG | Duo | Bla | Pro/Uth | mDice | Params | FLOPs[3] | PT score | mNSD |
|---|---|---|---|---|---|---|---|---|---|---|---|---|---|---|---|---|---|---|---|---|
| CoTr | 91.1 | 87.2 | 86.4 | 60.5 | 80.9 | 91.6 | 80.1 | 93.7 | 87.7 | 76.3 | 73.7 | 71.7 | 68.0 | 67.4 | 40.8 | 77.1 | 41.87 | 1510.53 | 1.07 | 64.2 |
| nnFormer | 95.9 | 93.5 | 94.8 | 78.5 | 81.1 | 95.9 | 89.4 | 94.2 | 88.2 | 85.0 | 75.0 | 75.9 | 78.5 | 83.9 | 74.6 | 85.6 | 150.14 | 1343.65 | 1.12 | 74.2 |
| UNETR | 92.7 | 88.5 | 90.6 | 66.5 | 73.3 | 94.1 | 78.7 | 91.4 | 84.0 | 74.5 | 68.2 | 65.3 | 62.4 | 77.4 | 67.5 | 78.3 | 93.02 | **391.03** | 1.41 | 61.5 |
| Swin UNETR | 95.5 | 93.8 | 94.5 | 77.3 | 83.0 | 96.0 | 88.9 | 94.7 | 89.6 | 84.9 | 77.2 | 78.3 | 78.6 | 85.8 | 77.4 | 86.4 | 62.83 | 1562.99 | 1.14 | 75.3 |
| VNet | 94.2 | 91.9 | 92.7 | 70.2 | 79.0 | 94.7 | 84.8 | 93.0 | 87.4 | 80.5 | 72.6 | 73.2 | 71.7 | 77.0 | 66.6 | 82.0 | 45.65 | 1737.57 | 1.10 | 67.9 |
| nnUNet[†] | **97.1** | 96.4 | 96.2 | 83.2 | 87.5 | 97.6 | 92.2 | **96.0** | 92.5 | 88.6 | 81.2 | 81.7 | **85.0** | 90.5 | 85.0 | 90.0 | 30.76 | 1067.89 | 1.30 | 82.1 |
| nnUNet[‡] | **97.1** | **97.0** | **97.1** | **86.6** | **87.7** | **97.9** | **92.4** | **96.0** | **92.7** | 88.8 | 81.6 | 82.1 | **85.0** | **90.6** | **85.2** | **90.5** | 30.76 | 1067.89 | 1.31 | **83.0** |
| E2ENet* (s=0.7) | **97.1** | 96.9 | **97.1** | 86.0 | 87.6 | **97.9** | 92.3 | 95.7 | 92.3 | **89.0** | 81.5 | **82.4** | 84.9 | 90.3 | 83.8 | 90.3 | 11.23 | 969.32 | 1.54 | 82.7 |
| E2ENet* (s=0.8) | **97.1** | 96.9 | 97.0 | 85.2 | 87.5 | **97.9** | 92.3 | 95.7 | 92.3 | **89.0** | 81.3 | 82.1 | 84.6 | 90.1 | 84.8 | 90.3 | 9.44 | 778.74 | 1.65 | 82.5 |
| E2ENet* (s=0.9) | 96.7 | 96.9 | 97.0 | 84.2 | 87.0 | 97.7 | 92.2 | 95.6 | 92.0 | 88.6 | 81.0 | 81.8 | 84.0 | 89.9 | 83.8 | 89.9 | 7.64 | 492.29 | **1.89** | 81.8 |
| E2ENet (s=0.7) | **97.1** | 96.6 | 96.5 | 83.4 | 87.6 | 97.5 | 92.3 | 95.8 | 92.3 | **89.0** | 81.4 | 82.3 | 84.9 | 90.3 | 83.8 | 90.1 | 11.23 | 969.32 | 1.54 | 82.3 |
| E2ENet (s=0.8) | **97.1** | 96.6 | 96.5 | 83.4 | 87.5 | 97.5 | 92.3 | 95.8 | 92.3 | **89.0** | 81.3 | 82.0 | 84.5 | 90.1 | 83.8 | 90.0 | 9.44 | 778.74 | 1.65 | 82.3 |
| E2ENet (s=0.9) | 96.7 | 95.4 | 96.4 | 82.6 | 86.9 | 97.4 | 92.2 | 95.6 | 92.0 | 88.6 | 80.9 | 81.7 | 84.0 | 89.9 | 83.8 | 89.6 | **7.64** | 492.29 | 1.88 | 81.4 |
| E2ENet(static, s=0.9) | 96.6 | 95.5 | 96.3 | 82.6 | 86.9 | 97.4 | 92.2 | 95.6 | 92.0 | 88.6 | 80.9 | 81.7 | 84.0 | 89.9 | 83.8 | 89.6 | **7.64** | 492.29 | 1.88 | 81.4 |

[3] The inference FLOPs are calculated based on the patch sizes of $1 \times 128 \times 128$ without considering postprocessing cost.

### A.8.2 BTCV Challenge

We compare the performance of our E2ENet model to several baselines (CoTr [Xie et al., 2021], RandomPatch [Tang et al., 2021], PaNN [Zhou et al., 2019], UNETR [Hatamizadeh et al., 2022], and nnUNet [Isensee et al., 2021]) on the test set of BTCV challenge, and report class-wise Dice, mDice, Params and inference FLOPs on the test set in Table 11. It is worth noting that nnUNet is a strong performer that uses an automatic model configuration strategy to select and ensemble

---

[8]https://github.com/google/XNNPACK.

[9]https://github.com/neuralmagic/deepsparse.

two best of multiple U-Net models (2D, 3D and 3D cascade) based on cross-validation results. In contrast, E2ENet is designed to be computationally and memory efficient, using a consistent 3D network configuration. Swin UNETR [Tang et al., 2022] is among the best on the leaderboard for this challenge. However, we do not include it in our comparison because it employs self-supervised learning with extra data. This falls outside of our goal of trading off training efficiency and accuracy without using extra data.

Our proposed E2ENet, a single 3D architecture without cascade, has achieved comparable performance to nnUNet, with mDice of 88.3%. Additionally, it has a significantly smaller number of parameters, 11.25 M, compared to other methods such as nnUNet (30.76 M), CoTr (41.87 M), and UNETR (92.78 M).

Table 11: Quantitative comparisons of segmentation performance on BTCV test set. Note: Spl: spleen, RKid: right kidney, LKid: left kidney, Gall: gallbladder, Eso: esophagus, Liv: liver, Sto: stomach, Aor: aorta IVC: inferior vena cava, Veins: portal and splenic veins, Pan: pancreas, AG: adrenal gland. The results (class-wise Dice and mDice) for these baselines are from [Hatamizadeh et al., 2022]. $^+$ denotes that the training of UNETR$^+$ is without using any extra data outside the challenge. The results of nnUNet$^‡$, E2ENet and Hausdorff Distance (HD)↓ of UNETR are from the standard leaderboard of BTCV challenge, while the results of nnUNet are from the free leaderboard.

| Methods | Spl | RKid | LKid | Gall | Eso | Liv | Sto | Aor | IVC | Veins | Pan | AG | mDice | Params | FLOPs [1] | PT score | HD |
|---|---|---|---|---|---|---|---|---|---|---|---|---|---|---|---|---|---|
| CoTr | 95.8 | 92.1 | 93.6 | 70.0 | 76.4 | 96.3 | 85.4 | 92.0 | 83.8 | 78.7 | 77.5 | 69.4 | 84.4 | 41.87 | 636.94 | 1.22 | / |
| RandomPatch | 96.3 | 91.2 | 92.1 | 74.9 | 76.0 | 96.2 | 87.0 | 88.9 | 84.6 | 78.6 | 76.2 | 71.2 | 84.4 | / | / | / | / |
| PaNN | 96.6 | **92.7** | 95.2 | 73.2 | 79.1 | 97.3 | 89.1 | 91.4 | 85.0 | 80.5 | 80.2 | 65.2 | 85.4 | / | / | / | / |
| UNETR$^+$ | 96.8 | 92.4 | 94.1 | 75.0 | 76.6 | 97.1 | 91.3 | 89.0 | 84.7 | 78.8 | 76.7 | 74.1 | 85.6 | 92.79 | **164.91** | 1.53 | 23.4 |
| nnUNet | **97.2** | 91.8 | **95.8** | 75.3 | 84.1 | **97.7** | **92.2** | 92.9 | **88.1** | 83.2 | **85.2** | 77.8 | **88.4** | 31.18 | 416.73 | 1.38 | 15.6 |
| nnUNet$^‡$ | 96.5 | 91.7 | **95.8** | 78.5 | 84.2 | 97.4 | 91.5 | 92.3 | 86.9 | 83.1 | 84.9 | 77.5 | 88.0 | 31.18 | 416.73 | 1.38 | 16.9 |
| E2ENet ($s=0.7$) | 96.5 | 91.3 | 95.7 | 78.1 | 84.5 | 97.5 | 91.5 | 92.2 | 86.7 | **83.4** | 84.8 | **77.9** | 88.3 | **11.25** | 449.00 | **1.68** | 16.1 |

$^1$ The inference FLOPs are calculated based on the patch sizes of $1 \times 96 \times 96 \times 96$. The codes for RandomPatch and PaNN are not publicly available, so it is not possible for us to determine their model size and inference FLOPs.

### A.8.3 Statistical Significance of Designed Modules

To demonstrate the advantages of individual modules, we plot a critical distance diagram using the Nemenyi post-hoc test with a p-value of 0.05 to establish the statistical significance of our modules. In Figure 7, the top line represents the axis along which the methods' average ranks, and a lower value indicates better performance. Methods joined by thick horizontal black lines are considered not statistically different.

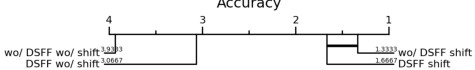

Figure 7: The critical distance diagram on the AMOS-CT validation dataset, with the evaluation metric being mDice.

From the diagram, we can clearly observe that E2ENet with depth shift significantly outperforms E2ENet without depth shift. Additionally, the incorporation of dynamic sparse feature fusion into E2ENet results in a substantial reduction in both the number of FLOPs (from 23.90M to 11.23M) and parameters (from 3069.55G to 969.32G) while maintaining comparable performance, without any significant performance degradation.

### A.9 The Impact of Weighting Factors $\alpha$ on PT Score

In this section, we investigate the impact of the weighting factors $\alpha_1$ and $\alpha_2$ on the Performance Trade-off Score for the AMOS-CT challenge, as defined in Equation 7. These factors are used to balance the trade-off between accuracy and resource cost. A higher value of $\alpha_1$ prioritizes accuracy, while a lower value emphasizes resource cost. Figure 8 shows that as $\alpha_1$ decreases, the gaps in the Performance Trade-off Score between E2ENet and other methods become larger, indicating that our method is more advantageous when prioritizing resource cost. However, even when $\alpha_1$ is set to be 20 times greater than $\alpha_2$, which prioritizes accuracy over resource cost, the Performance Trade-off Score of E2ENet remains superior to other baselines. This result indicates that our proposed E2ENet architecture is highly efficient in terms of computational cost and memory usage while achieving excellent segmentation performance on the AMOS-CT challenge.

### A.9.1 Qualitative Results

**BTCV Challenge** In Figure 9 (b), we present a qualitative comparison of our proposed E2ENet method with nnUNet as a baseline model on the BTCV challenge. Our results demonstrate the

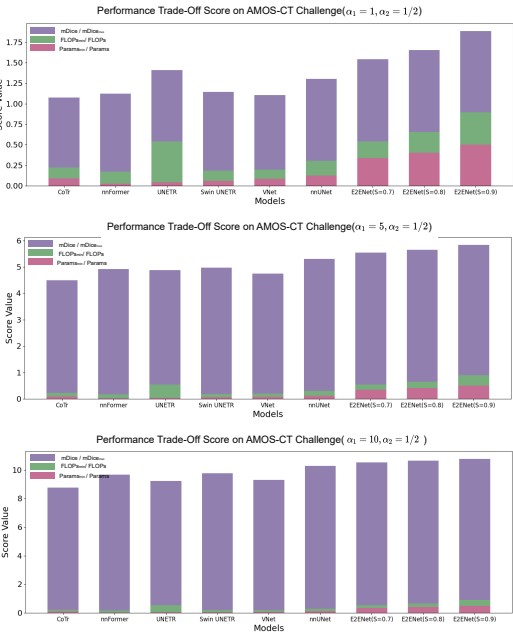

Figure 8: Comparison of Performance Trade-Off score between E2ENet and other models on AMOS-CT challenge with varying $\alpha_1$ and $\alpha_2$ values. E2ENet outperforms other baselines in achieving a better trade-off between accuracy and efficiency across different preferences.

effectiveness of our proposed method in addressing some of the challenges of medical image segmentation. For example, as shown in the first and third columns, our E2ENet method accurately distinguishes the stomach from the background without over- or under-segmentation, which can be difficult due to the low contrast in the image. In the second column, E2ENet performs well in differentiating the stomach from the spleen. These examples suggest that our DSFF module can effectively encode feature information for improved performance in medical image segmentation.

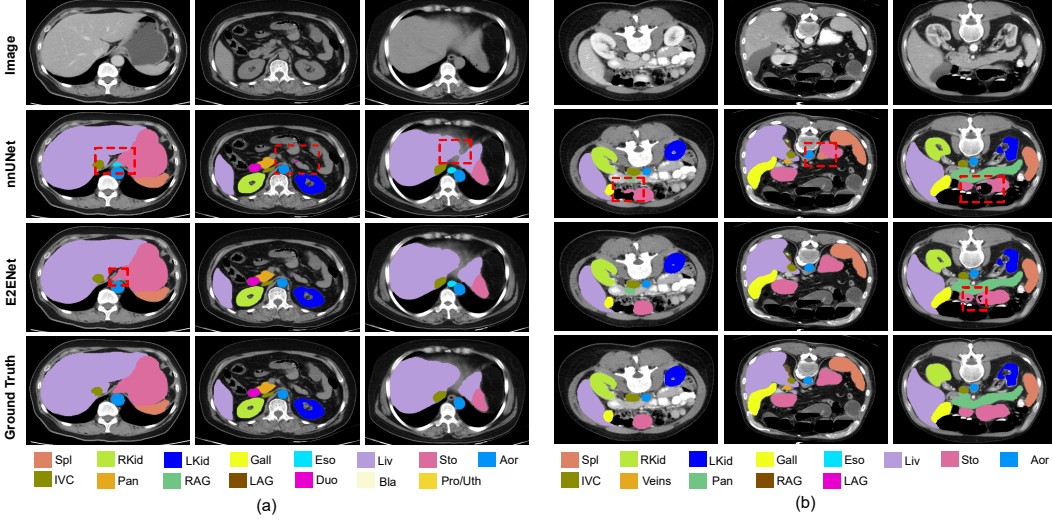

Figure 9: Qualitative comparison of the proposed E2ENet and nnUNet on AMOS-CT and BTCV challenges.

**BraTS Challenge in MSD** Figure 10 presents a qualitative comparison of our proposed E2ENet method with the nnUNet on the BraTS challenge with highly variable shapes of the segmentation

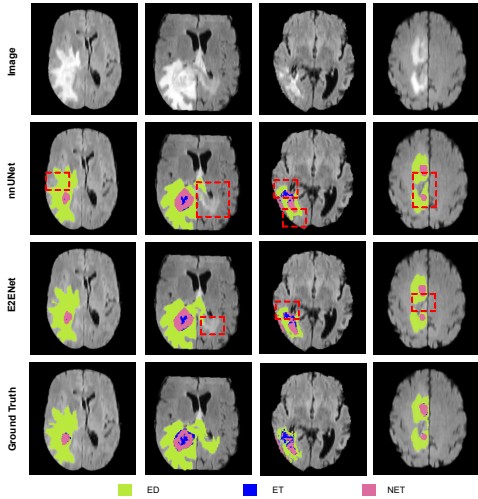

Figure 10: Qualitative comparison of the proposed E2ENet and nnUNet on BraTS Challenge in MSD.

targets. Based on the results of the baseline model, nnUNet, we observed that accurately distinguishing the edema (ED) from the background is difficult, as the edema tends to have less smooth boundaries. Our results suggest that E2ENet may have some potential to improve the distinguishability of the edema boundaries, as evidenced by the relatively better segmentation results in the first, second, and fourth columns. Moreover, E2ENet accurately differentiates the enhanced tumor (ET) from the edema, as shown in the third column, which is a challenging task due to the similarity in appearance between these two regions, and the dispersive distribution of ET. These findings suggest that E2ENet is a promising method for accurately segmenting brain tumors in challenging scenarios.

### A.10 Convergence Analysis

In this section, we analyze the convergence behavior of E2ENet by examining the loss changes during topology updating (kernel activation/deactivation epochs), comparing it with the best-performing baseline nnUNet, and studying the impact of topology update frequency. From Figure 11, we observed that the activation/deactivation of weights initially led to an increase in training loss. However, over the long term, the training converged. Additionally, we compared the learning curve of E2ENet with that of nnUNet and found that E2ENet converged even faster than nnUNet, as shown in the subplot in Figure 12 (a). To account for the effect of the number of parameters, we scaled down nnUNet to have a similar number of parameters as E2ENet and observed that it converged even more slowly than the original nnUNet. We also studied the impact of topology update frequency. As shown in Figure 12 (b), when the topology updating frequency is increased, the convergence speed may decrease slightly, but the impact is not significant.

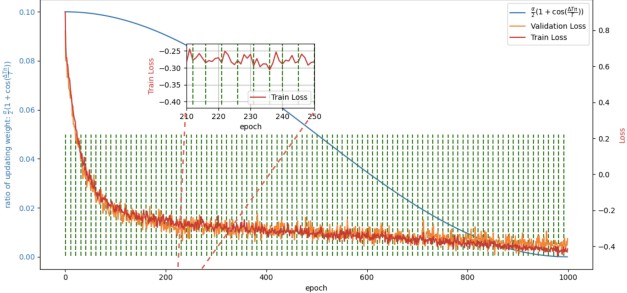

Figure 11: The learning curve of E2ENet on AMOS-CT, with green dotted vertical lines indicating the epochs of weight activation and deactivation. The blue line represents the ratio of weight deactivation/reactivation throughout the training process.

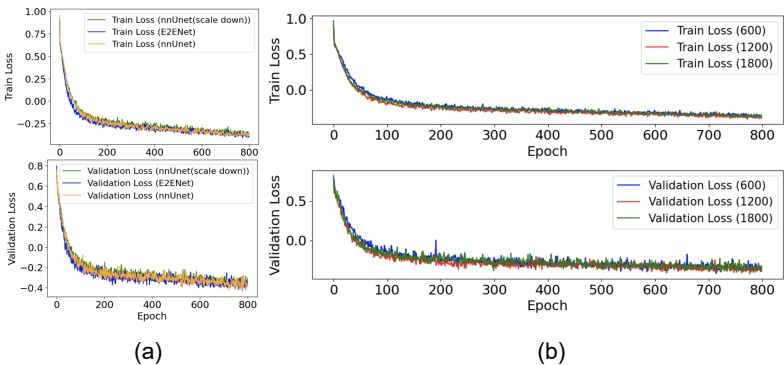

(a)                                          (b)

Figure 12: (a) Comparing the learning curve of E2ENet with that of nnUNet and scaled-down nnUNet (referred to as nnUNet (-)); (b) Comparing the learning curve of E2ENet with different topology update frequencies.

## A.11   Organ Volume Statistics and Class-wise Results Visualization

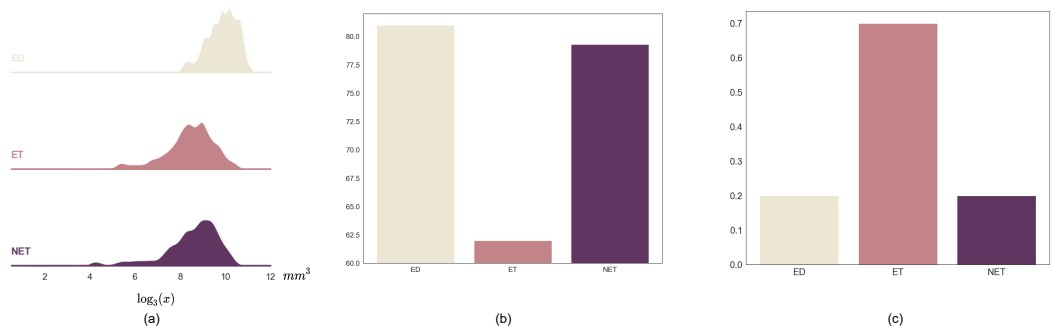

(a)                                    (b)                                    (c)

Figure 13: (a) The organ volume statistics of AMOS-CT training dataset. (b) Class-wise Dice of nnUNet without postprocessing (visualization of Table 1). (c) Class-wise Dice differences between E2ENet with feature sparsity 0.7 without postprocessing and nnUNet without postprocessing on AMOS-CT validation dataset. The positive value means that E2ENet outperforms nnUNet, and vice versa.)

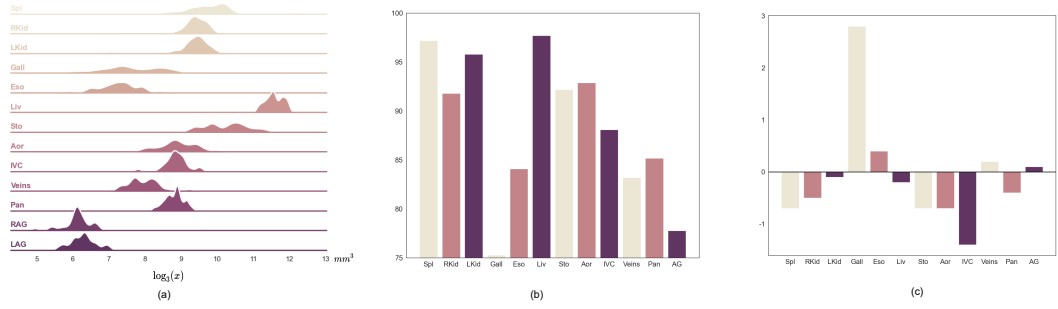

(a)                                    (b)                                    (c)

Figure 14: (a) The organ volume statistics of BTCV training dataset. (b) Class-wise Dice of nnUNet (visualization of Table 11). Note that AG denotes the average of the right and left adrenal glands (RAG and LAG). (c) Class-wise Dice differences between E2ENet with feature sparsity 0.7 and nnUNet on BTCV test dataset. The positive value means that E2ENet outperforms nnUNet, and vice versa.

In this section, we analyzed the relationship between organ volume and segmentation accuracy on the AMOS-CT, BTCV, and BraTS challenges. The results, depicted in Figures 13, 14 and 15, showed that small organs with relatively low segmentation accuracy. For the AMOS-CT challenge, RAG

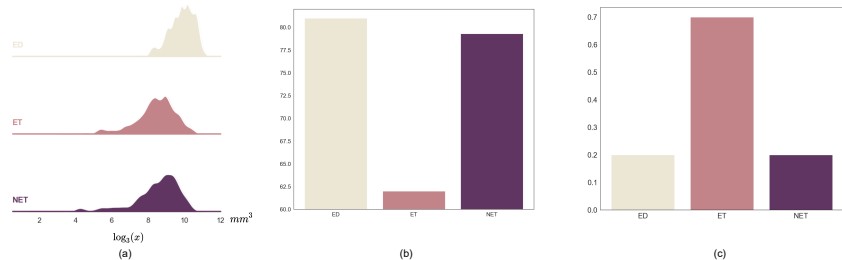

Figure 15: (a) The organ volume statistics of BraTS training dataset. (b) Class-wise Dice of nnUNet (visualization of Table 2) (c) Class-wise Dice differences between E2ENet with feature sparsity 0.7 and nnUNet on 5-fold cross-validation of the training dataset. The positive value means that E2ENet outperforms nnUNet, and vice versa.

(right adrenal gland), LAG (left adrenal gland), Gall (gallbladder), and Eso (esophagus) are more challenging to accurately segment. This may be due to the fact that smaller organ volumes provide less visual information for the segmentation algorithm to work with. However, our proposed method, E2ENet, also demonstrated comparable (or better) performance on these small organs, particularly for the organ "LAG", in which the Dice improved from $81.7\%$ to $82.4\%$. On the BTCV challenge, the Dice of "Gall", which is considered to be the most challenging organ, improves from $75.3\%$ to $78.1\%$ when using E2ENet compared to nnUNet. For the BraTs challenge, E2ENet demonstrates the most significant improvement in the Dice score of the "ET" region, which is considered the most challenging class, with an increase of $0.7\%$.

These results indicate that by applying the DSFF mechanism, E2ENet is able to effectively utilize multi-scale information, potentially leading to improved performance in segmenting small organs. It is important to note that other factors, such as the quality and resolution of the medical images, as well as the complexity of the anatomy being imaged, may also impact the performance of the segmentation algorithms. Future work could focus on further exploring the potential impact of these factors on segmentation accuracy.

