# OpenReview forum: "E2ENet: Dynamic Sparse Feature Fusion for Accurate and Efficient 3D Medical Image Segmentation"
_NeurIPS.cc/2024/Conference — NeurIPS 2024 poster_

### Official Review · Reviewer_SYnH · 2024-06-22

**Soundness:** 3
**Presentation:** 3
**Contribution:** 3
**Rating:** 6
**Confidence:** 3

**Summary:**

This paper introduces E2ENet, a 3D medical image segmentation model designed for efficiency and performance. E2ENet incorporates Dynamic Sparse Feature Fusion (DSFF) to adaptively fuse informative multi-scale features and a Restricted Depth-Shift mechanism in 3D convolution to maintain low model complexity. Extensive experiments demonstrate that E2ENet achieves a superior trade-off between accuracy and efficiency, significantly reducing parameter count and FLOPs compared to previous methods, particularly in large-scale datasets like AMOS-CT.

**Strengths:**

Improved Efficiency: E2ENet significantly reduces the parameter count and FLOPs, making it more computationally efficient and suitable for deployment on resource-limited hardware without compromising on performance.
Innovative Mechanisms: The introduction of Dynamic Sparse Feature Fusion (DSFF) and Restricted Depth-Shift in 3D convolution effectively balances the need for high accuracy with lower computational complexity, offering a novel approach to 3D medical image segmentation.
Robust Validation: Extensive experiments on multiple challenging datasets demonstrate E2ENet's consistent performance, ensuring its reliability and applicability in various medical imaging scenarios, especially for exceeding nnUNet.

**Weaknesses:**

1. Unclear Backbone Network: The backbone network of E2ENet is not clearly defined. Figure 2 labels the left part as an "efficient backbone," but it is ambiguous whether this refers to an EfficientNet-based backbone or simply a group of CNN layers. This lack of clarity can confuse readers and detract from the paper's overall comprehensibility.
2. Segmentation Performance and Kernel Size: The segmentation performance is primarily evaluated on the BraTS and AMOS datasets. However, the ablation study on AMOS does not adequately address the suitability of the DSFF kernel size for BraTS. Table 4 lacks results for a kernel size of 3x3x3, which raises questions about the generalizability of the chosen kernel sizes across different datasets.
3. Lack of Comparison with Recent Lightweight Networks: The paper does not include comparisons with recent lightweight network structures specifically designed for medical image analysis. This omission limits the assessment of how E2ENet stands relative to other contemporary, efficient models and reduces the comprehensiveness of the evaluation.

**Questions:**

1. Clarification on the Backbone Network:
Question: Can you provide a more detailed description of the backbone network used in E2ENet? Specifically, is it an EfficientNet-based backbone, or is it composed of a different set of CNN layers?
Suggestion: Consider revising Figure 2 and the corresponding text to clearly define the architecture of the backbone network. Providing explicit details will enhance the readers' understanding and reduce ambiguity.
2. Kernel Size Generalization:
Question: Why was the 3x3x3 kernel size not included in the ablation study results for the BraTS dataset in Table 4?
Suggestion: It would be beneficial to include the results for the 3x3x3 kernel size in the ablation study for the BraTS dataset. This would help in understanding the generalizability of the DSFF mechanism across different datasets and ensure that the chosen kernel sizes are suitable for various segmentation tasks.
3. Comparison with Recent Lightweight Networks:
Question: Have you considered comparing E2ENet with other recent lightweight network structures specifically designed for medical image analysis?
Suggestion: Including comparisons with recent lightweight networks would strengthen the paper by providing a more comprehensive evaluation of E2ENet's performance. This could involve benchmarking against models that are known for their efficiency and effectiveness in medical image segmentation.

---

> ### Author Rebuttal · Authors · 2024-08-06
>
> ## To Reviewer SYnH
>
> We would like to thank the reviewer for your thoughtful and detailed comments. We are glad that you appreciate the efficiency improvements and robust validation presented in our paper. We address your comments below.
>
> > Unclear Backbone Network: The backbone network of E2ENet is not clearly defined. Figure 2 labels the left part as an "efficient backbone," but it is ambiguous whether this refers to an EfficientNet-based backbone or simply a group of CNN layers. This lack of clarity can confuse readers and detract from the paper's overall comprehensibility.
>
> Thank you for pointing out the description of the backbone network, which indeed helps to enhance the readers' understanding and reduce ambiguity. We will provide this information below and add it to our paper.
>
> The efficient backbone network consists of several levels as in Figure 2, each comprising two consecutive blocks. Each block includes our proposed efficient Restricted Depth-Shift 3D Convolutional layer, followed by instance normalization and ReLU activation (termed **conv–norm–relu**). After each level, the downsampling is performed using a strided convolution operation in the second block of that level (the convolution in the second block of the new resolution has a stride >1).
> Table: Backbone Network – Each row describes level $i$, with input resolution $H^i$, $W^i$, $D^i$, and output channels $C^i$, given an input size of 128×128×128.
>
> | level $i$ | Operator | Resolution $H^i$, $W^i$, $D^i$  |  Channels $C^i$   |
> | -------- | -------- | -------- | --- |
> | 1     | conv–norm–relu + conv–norm–relu      | 128x128x128     |  48  |
> | 2     | conv–norm–relu + conv–norm–relu      | 64x64x64     |  96  |
> | 3     | conv–norm–relu + conv–norm–relu      | 32x32x32     |  192  |
> | 4     | conv–norm–relu + conv–norm–relu      | 16x16x16     |  320  |
> | 5     | conv–norm–relu + conv–norm–relu      | 8x8x8     |  320  |
>
> > Kernel Size Generalization: Question: Why was the 3x3x3 kernel size not included in the ablation study results for the BraTS dataset in Table 4? Suggestion: It would be beneficial to include the results for the 3x3x3 kernel size in the ablation study for the BraTS dataset.
>
> Thank you for your valuable suggestion. Indeed, a 3x3x3 kernel size is helpful in the ablation study for the BraTS dataset as well. This will aid in understanding the generalizability of the DSFF mechanism across different datasets. Here, we update these results as follows and include them in Table 4 of the paper:
>
> | w/ DSFF | shift | kernel size | ED  | ET  | NET | mDice | Params | FLOPs |
> | ------- | ----- | ----------- | --- | --- | --- | ----- | ------ | ----- |
> |   No |    No   |      3x3x3   |  80.9   | 61.9    |   79.1  |   74.0    |   52.55     | 4519.26|
> |   Yes    |   No    |    3x3x3 | 81.0    | 62.2    |  79.1  |  74.1     |   28.02     | 2023.52 |
> |   No |    Yes   |      1x3x3   |  81.0    |  62.3   |  79.0   |   74.1    | 23.89 |  3071.78|
> |   Yes    |   Yes    |  1x3x3 |   **81.2**  |  **62.7**   |  **79.5**    |   **74.5**     | 11.24| 1067.06|
>
> We find that, compared to kernel sizes of 3x3x3, E2ENet with kernel sizes of 1x3x3 combined with a depth shift (3rd and 4th row) maintains or even improves segmentation accuracy, whether with DSFF or without DSFF. This further demonstrates that our **proposed efficient Restricted Depth-Shift 3D Convolutional layer, which utilizes a 1x3x3 kernel with restricted depth shift, is equivalent to a 3x3x3 kernel in terms of segmentation accuracy.** Moreover, it offers significant **savings in computational and memory resources**, as observed in the AMOS-CT dataset in Table 3.
>
> > Comparison with Recent Lightweight Networks: Question: Have you considered comparing E2ENet with other recent lightweight network structures specifically designed for medical image analysis? Suggestion: Including comparisons with recent lightweight networks would strengthen the paper by providing a more comprehensive evaluation of E2ENet's performance.
>
> Thank you for your suggestion. We agree that comparisons with recent lightweight networks would strengthen the paper. We have included another recent efficient lightweight network, UNETR++ [1], as our baseline. UNETR++ offers both high-quality segmentation accuracy and efficiency. From the results below, we can see that E2ENet achieves better mDice scores. This further verifies the efficiency and accuracy of our proposed model for 3D medical segmentation.
>
> Moreover, we conducted an extended ablation study by integrating our proposed DSFF into UNETR++[1], referred to as **UNETR++ w/ DSFF** in the table below. Specifically, we introduced dynamic sparsity into UNETR++ by sparsifying its connections: **less important activated connections are removed, while the same number of deactivated connections are randomly reactivated during training**. The results show that parameters can be further reduced while maintaining stable mDice performance. This further demonstrates the effectiveness of our proposed DSFF, showing that it can be **easily integrated into recent lightweight networks to potentially enhance efficiency while maintaining mDice performance.**
>
> | model | ED  | ET  | NET | mDice | Params
> ----------- | --- | --- | --- | ----- | ------ |
> |  UNETR++ |   80.2   |  61.0  |   78.7  |  73.3   |  33.8 |
> | UNETR++ w/ DSFF | 80.4    | 61.4  |  78.4   | 73.4 | 5.11 |
> |  E2ENet   |   81.2  |  62.7   |  79.5    |   74.5  | 11.24|
>
> [1] Abdelrahman Shaker, Muhammad Maaz, Hanoona Rasheed, Salman Khan, Ming-Hsuan Yang and Fahad Shahbaz Khan. UNETR++: Delving into Efficient and Accurate 3D Medical Image Segmentation. IEEE Transactions on Medical Imaging, 2024.
>
> We thank you again for the time and effort you've taken to participate in the review of our paper. If you have further questions and concerns, we are more than happy to discuss with you.

---

> > ### Comment · Reviewer_SYnH · 2024-08-11
> > **Response**
> >
> > Thank you for addressing my concerns, i would raise my score to week accept.

---

> ### Author Response · Authors · 2024-08-11
> **Thank you for your support!**
>
> Dear Reviewer SYnH,
>
> We sincerely thank you for your constructive comments and support!
>
> Best,
>
> Authors

---

### Official Review · Reviewer_JUDh · 2024-07-09

**Soundness:** 4
**Presentation:** 3
**Contribution:** 3
**Rating:** 6
**Confidence:** 3

**Summary:**

The paper introduces E2ENet, a novel neural network designed for 3D medical image segmentation, which emphasizes efficiency in computational resource usage without compromising accuracy. This paper introduces a Dynamic Sparse Feature Fusion (DSFF) mechanism that adaptively learns to integrate multi-scale features effectively and a novel application of restricted depth-shift in 3D convolution that aligns with the computational simplicity of 2D methods. The model demonstrates superior performance on various benchmarks like AMOS-CT challenge and BraTS Challenge in MSD, showcasing significant reductions in parameters and computational costs while maintaining competitive accuracy.

**Strengths:**

(1) DFSS mechanism proposed provides a more efficient feature fusion process while reducing the computational and memory overhead. (2) E2ENet integrates depth-shift strategy in 3D convolution networks, enabling the ability for network to capture 3D spatial relationships. (3) E2ENet significantly reduces parameter size to 7.63 M minimally.

**Weaknesses:**

Section 3.2 Ablation Studies lacks of more insights about Table 3, detail question will be shown in Question part below.

**Questions:**

For Section 3.2 Table 3, it seems that for the different shift size, the mDice score does not vary much from each other. What are authors’ insights about this? Based on this, how do authors justify in Section 2.3 that the shift magnitude would have a negative impact on the effectiveness of the shift operation?

**Limitations:**

Yes, authors have addressed the limitations.

---

> ### Author Rebuttal · Authors · 2024-08-06
>
> ## To Reviewer JUDh
>
> Thank you very much for taking the time to review our paper and for your helpful comments. We provide detailed responses to your constructive feedback below.
>
> > **For Section 3.2 Table 3, it seems that for the different shift size, the mDice score does not vary much from each other. What are authors’ insights about this?**
>
>
> Thank you for your enthusiasm for our paper and for sharing your concerns with us. Indeed, we observe a marginal difference in mDice, with a decrease of 0.4 when the shift size magnitude is increased from 1 to 3. However, for mNSD, there is a significant performance drop from 82.3 to 81.6. This is because the mDice score primarily measures the **overlap between the predicted segmentation and the ground truth**, whereas mNSD focuses on the **distance between the surfaces (boundaries)**. This demonstrates that shift size is more impactful on boundary alignment.
>
> Furthermore, in additional experiments where we extended the shift size magnitude to 7, we observed a further decrease in performance by 2.5 for mDice and 4.8 for mNSD, compared to a shift size magnitude of 1. The detailed results are shown below. In summary, the shift operation is beneficial, resulting in a 1.9 improvement in mDice compared to not using the shift operation. However, our experiment results indicate that the magnitude should not be set too large. We will explain more below.
>
>
> | w/ shift | shift size | kernel size  | mDice  |mNSD |
> | ----- | ----------- | --- | --- | --- |
> |   No    |   --  | 1x3x3    |  88.6   |   78.6 |
> |   Yes    |   (−1, 0, 1)  | 1x3x3    |  **90.1**   |   **82.3** |
> |   Yes    |   (−2, 0, 2)  | 1x3x3    |   89.8  | 82.0     |
> |   Yes    |   (−3, 0, 3)  | 1x3x3    |  89.7   |   81.6   |
> |   Yes    |   (−7, 0, 7)  | 1x3x3    |   87.6  |   77.5   |
>
>
> > **How do authors justify in Section 2.3 that the shift magnitude would have a negative impact on the effectiveness of the shift operation?**
>
> In Section 2.3, we propose the Restricted Depth-Shift operation, which aims to capture depth-wise information while maintaining a 2D computation cost. However, increasing the shift size means considering more depth-wise information **at the expense of channel-wise information**, leading to an insufficient representation of channels, as discussed in Section 3. Additionally, a large shift size causes a **loss of local spatial relationships**, which are crucial for segmentation. This results in a blurring effect that reduces the precision of boundary alignment, particularly affecting metrics like mNSD, which rely heavily on accurate boundary information.
>
> We would like to thank you again for your time and effort. If you have further questions, we are more than happy to discuss them with you.

---

> > ### Author Response · Authors · 2024-08-13
> >
> > Dear reviewer JUDh, we appreciate your valuable feedback and constructive comments. Since there is only one day left in the rebuttal process, we want to know if our response addresses your concerns.

---

> > > ### Comment · Reviewer_JUDh · 2024-08-13
> > >
> > > Authors, thank you so much for your detailed response. I will keep my score.

---

> > > > ### Author Response · Authors · 2024-08-13
> > > >
> > > > Dear Reviewer JUDh,
> > > >
> > > > We sincerely thank you for your response and positive support!
> > > >
> > > > Best,
> > > > Authors

---

### Official Review · Reviewer_xwUb · 2024-07-11

**Soundness:** 3
**Presentation:** 3
**Contribution:** 3
**Rating:** 6
**Confidence:** 4

**Summary:**

In this paper, the authors propose a novel architecture that addresses the challenges observed in increasing the model size and computational complexity of neural network architectures. This leads to concerns in the deployment stage, mainly because of resource-limited hardware. The authors propose a 3D medical image segmentation model named Efficient to Efficient Network (E2ENet). They incorporated two designs to make the model efficient while preserving accuracy: Dynamic sparse feature fusion (DSFF) mechanism and Restricted depth-shift in 3D convolution.  Extensive experiments on three benchmarks show that E2ENet consistently achieves a superior trade-off between accuracy and efficiency compared to prior state-of-the-art baselines.

**Strengths:**

The paper is well-organized and well-written.
The paper reads well.
The motivation behind the study is clear.
The proposed restricted depth shift method is interesting and somewhat novel.

**Weaknesses:**

Tables 3 and 4 don’t provide consistent results. It seems that the combination that works for CT does not optimally work for MRIs.
Missing discussion on limitations.

**Questions:**

Can the proposed method accelerate training speed in terms of time to achieve SOTA accuracies?
Could the authors provide information on the inference times?

**Limitations:**

The authors didn't discuss the limitations of the proposed methodology.

---

> ### Author Rebuttal · Authors · 2024-08-06
>
> ## To Reviewer xwUb
>
> We sincerely thank you for your time and effort in reviewing our paper. We are glad that you find our work interesting and novel. We address your comments below.
>
> > Tables 3 and 4 don’t provide consistent results. It seems that the combination that works for CT does not optimally work for MRIs.
>
> For the mDice score, in Table 3, the shift operation in E2ENet with and without DSFF combination scores 90.1 and 90.2, respectively. In Table 4, with and without DSFF scores 81.2 and 81.0, respectively. These results rank among the top in these tables and show marginal differences in performance.
>
> However, with the DSFF combination, while it does not significantly improve the mDice scores, the computational cost in terms of parameters is notably reduced from 23.9M to 11.2M, a **2-fold reduction** for both Table 3 and Table 4.
>
> - Performance: The mDice scores show marginal differences, indicating that the shift operation combined with DSFF **maintains comparable performance** to the shift operation without DSFF.
> - Efficiency: The incorporation of DSFF into E2ENet results in a substantial **reduction in the number of parameters and FLOPs**, achieving a significant computational cost reduction.
> - Conclusion: As discussed in Section 3.2, our main conclusion is that dynamic sparse feature fusion significantly reduces computational resources while maintaining high performance without any significant degradation.
>
> Furthermore, in Appendix B.7.3, we provide additional verification on the **statistical significance of the designed modules: DSFF and depth-shift operation**.
>
>
> > Can the proposed method accelerate training speed in terms of time to achieve SOTA accuracies? Could the authors provide information on the inference times?
>
> Thank you for your enthusiasm for our paper and for sharing your concerns with us.
> Given the relatively restricted support for sparse operations in current off-the-shelf commodity GPUs and TPUs without sparsity-aware accelerators, we did not attempt to achieve practical speedup during training. Instead, we chose to implement our models with binary masks in our work. As mentioned extensively in the conclusion and appendix, the promising benefits of dynamic sparsity presented in this study have not yet translated into actual speedup. Accelerating training time will be a focus for our next work.
>
> Although not the focus of our current work, it would be interesting for future work to examine the speedup results of sparse operation during training, using such specialized hardware accelerators, as we see much improvement room of promise here. For example, at high unstructured sparsity levels, XNNPACK (https://github.com/google/XNNPACK) has already shown significant speedups over dense baselines on smartphone processors.
>
>
> **For the inference time:**
>
> Although the support for unstructured sparsity on GPUs remains relatively limited, its practical relevance has been widely demonstrated on non-GPU hardware, such as CPUs and customized accelerators. For instance, FPGA accelerators for an unstructured sparse RNN achieved high acceleration and energy efficiency compared to commercial CPUs and GPUs by maximizing the use of embedded multiply resources on the FPGA. Another notable success is DeepSparse (https://github.com/neuralmagic/deepsparse), which successfully deploys large-scale BERT-level sparse models on modern Intel CPUs, achieving a 10× model size compression with less than 1% accuracy drop, a 10× CPU-inference speedup with less than 2% accuracy drop, and a 29× CPU-inference speedup with less than 7.5% accuracy drop.
>
> Inspired by these advancements, we adopted an approach based on DeepSparse. We conducted experiments with patches of images sized 32×32×32 as input, comparing the CPU wall-clock timings for online inference between our proposed E2ENet and nnUNet on an Intel Xeon Platinum 8360Y CPU with 18 cores. We acknowledge that, while **our proposed models with sparsity do achieve speedups in practical inference**, they are not as pronounced as those observed with BERT-level sparse models. This is primarily due to the nature of segmentation and 3D convolution operations. However, this presents a promising avenue for our future work.
>
>
> | Methods | nnUNet | E2ENet(s=0.8) | E2ENet(s=0.9)|
> | -------- | -------- | -------- | --- |
> | Latency(ms)     | 10.07     | 8.02     |  **7.28**  |
> | Throughput(items/sec)     | 99.19     | 124.60     |  **137.13**   |
> | speedup     | 1.0x | 1.26x | **1.38x** |
>
>
> > Missing discussion on limitations.
>
> We appreciate the reviewer pointing this out. I will make it more clear in the final version. Our proposed model leverages unstructured dynamic sparsity; however, due to the relatively restricted support for unstructured sparsity in current off-the-shelf commodity GPUs, we did not attempt to achieve practical speedup during training on GPU. This limitation presents a potential direction for our future work.
>
> We thank you again for the time and effort you've taken to participate in the review of our paper. If you have further questions, we are more than happy to discuss with you.

---

> > ### Comment · Reviewer_xwUb · 2024-08-13
> >
> > Thank you, authors, for the detailed response feedback. After carefully considering your comments, I believe that the initial score accurately reflects the strengths of this work.

---

> > > ### Author Response · Authors · 2024-08-13
> > >
> > > Dear Reviewer xwUb, we appreciate your constructive comments and feedback! Thank you for your positive support!

---

### Author Rebuttal · Authors · 2024-08-06

## To All Reviewers:

We thank the reviewers for their constructive suggestions and in-depth analysis, which is helpful for our work. We are humbled by such a positive response, and we truly appreciate it.

We are delighted to note that all reviewers recognized the novelty of our research, found the idea **well-motivated**, **interesting**, and **novel** (Reviewers xwUb & SYnH), appreciated our contributions towards **improving efficiency** (Reviewers JUDh & SYnH), and acknowledged the **robust validation** (Reviewer SYnH). Additionally, the reviewer xwUb found our work **well-organized** and **well-written**.

In response to the reviewers' valuable feedbacks, we have made earnest efforts to address all raised concerns. A summary of our responses is as follows:

- Explained the results in Tables 3 and 4. (Reviewers xwUb)
- Reported the inference time. (Reviewers xwUb)
- Provided more insight for Table 3. (Reviewers JUDh)
- Explained our backbone (Reviewer SYnH)
- Added the results for kernel size of 3x3x3 on the BraTS dataset (Reviewer SYnH)
- Compared with recent lightweight networks (Reviewer SYnH)

Should there be any points that remain unclear or require further clarification, please do not hesitate to bring them to our attention. We are open to any additional feedback, comments, or suggestions, and we sincerely appreciate your continued engagement in enhancing the quality of our work.

---

### Comment · Area_Chair_adQN · 2024-08-11
**Author-reviewer discussion**

Dear reviewers,

Since the authors provided their responses, please read the responses, respond to them on in the discussion, and discuss points of disagreement if necessary by Aug 13.

Best regards,

AC

---

### Decision · Program_Chairs · 2024-09-25

**Decision:**

Accept (poster)

**Comment:**

This paper receives all positive reviews. Reviewers recognize the novelty of the paper and enjoy the efficiency of the proposed method. The AC suggests acceptance and encourages the authors to include the discussion in the rebuttal into the final version.